# EphrinB2-EphB4 signalling provides Rho-mediated homeostatic control of lymphatic endothelial cell junction integrity

Maike Frye[1,2]*, Simon Stritt[1], Henrik Ortsäter[1], Magda Hernandez Vasquez[1], Mika Kaakinen[3], Andres Vicente[4†], John Wiseman[5], Lauri Eklund[3,6], Jorge L Martínez-Torrecuadrada[7], Dietmar Vestweber[8], Taija Mäkinen[1]*

[1]Uppsala University, Department of Immunology, Genetics and Pathology, Uppsala, Sweden; [2]University Medical Center Hamburg-Eppendorf, Institute of Clinical Chemistry and Laboratory Medicine, Hamburg, Germany; [3]Biocenter Oulu, University of Oulu, Oulu, Finland; [4]Lymphatic Development Laboratory, Cancer Research UK London Research Institute, London, United Kingdom; [5]Discovery Biology, Discovery Sciences, Biopharmaceuticals R&D, AstraZeneca, Gothenburg, Sweden; [6]Oulu Centre for Cell-Matrix Research, Faculty of Biochemistry and Molecular Medicine, University of Oulu, Oulu, Finland; [7]Crystallography and Protein Engineering Unit, Spanish National Cancer Research Centre (CNIO), Madrid, Spain; [8]Max Planck Institute for Molecular Biomedicine, Münster, Germany

*For correspondence:
m.frye@uke.de (MF);
taija.makinen@igp.uu.se (TM)

Present address: †Institute for Global Prosperity, University College London, London, United Kingdom

**Abstract** Endothelial integrity is vital for homeostasis and adjusted to tissue demands. Although fluid uptake by lymphatic capillaries is a critical attribute of the lymphatic vasculature, the barrier function of collecting lymphatic vessels is also important by ensuring efficient fluid drainage as well as lymph node delivery of antigens and immune cells. Here, we identified the transmembrane ligand EphrinB2 and its receptor EphB4 as critical homeostatic regulators of collecting lymphatic vessel integrity. Conditional gene deletion in mice revealed that EphrinB2/EphB4 signalling is dispensable for blood endothelial barrier function, but required for stabilization of lymphatic endothelial cell (LEC) junctions in different organs of juvenile and adult mice. Studies in primary human LECs further showed that basal EphrinB2/EphB4 signalling controls junctional localisation of the tight junction protein CLDN5 and junction stability via Rac1/Rho-mediated regulation of cytoskeletal contractility. EphrinB2/EphB4 signalling therefore provides a potential therapeutic target to selectively modulate lymphatic vessel permeability and function.

## Introduction

The endothelium of blood and lymphatic vessels forms a barrier that controls the movement of fluid, molecules, ions and cells between the blood/lymph and the tissue. The barrier function of endothelial cells (ECs) varies among different organs and vessel types. The blood brain barrier (BBB), for example, is formed of a continuous layer of ECs connected by specialized tight junctions and adherens junctions (*Zhao et al., 2015*). In contrast, blood vessels of the kidney and small intestine are lined by fenestrated ECs to facilitate rapid exchange, uptake and secretion of fluids, solutes and molecules (*Aird, 2012*). The architecture of lymphatic endothelial cell (LEC) junctions also differs between the different vessel types (*Baluk et al., 2007*). Highly permeable button-like junctions of lymphatic capillaries allow uptake of fluid from the interstitium. The lymph is then drained and

**eLife digest** Lymph vessels are thin walled tubes that, similar to blood vessels, carry white blood cells, fluids and waste. Unlike veins and arteries, however, lymph vessels do not carry red blood cells and their main function is to remove excess fluid from tissues. The cells that line vessels in the body are called endothelial cells, and they are tightly linked together by proteins to control what goes into and comes out of the vessels. The chemical, physical and mechanical signals that control the junctions between endothelial cells are often the same in different vessel types, but their effects can vary.

The endothelial cells of both blood and lymph vessels have two interacting proteins on their membrane known as EphrinB2 and its receptor, EphB4. When these two proteins interact, the EphB4 receptor becomes activated, which leads to changes in the junctions that link endothelial cells together. Frye et al. examined the role of EphrinB2 and EphB4 in the lymphatic system of mice. When either EphrinB2 or EphB4 are genetically removed in newborn or adult mice, lymph vessels become disrupted, but no significant effect is observed on blood vessels. The reason for the different responses in blood and lymph vessels is unknown.

The results further showed that lymphatic endothelial cells need EphB4 and EphrinB2 to be constantly interacting to maintain the integrity of the lymph vessels. Further examination of human endothelial cells grown in the laboratory revealed that this constant signalling controls the internal protein scaffold that determines a cell's shape and integrity. Changes in the internal scaffold affect the organization of the junctions that link neighboring lymphatic endothelial cells together.

The loss of signalling between EphrinB2 and EphB4 in lymph vessels reflects the increase in vessel leakage seen in response to bacterial infections and in some genetic conditions such as lymphoedema. Finding ways to control the signalling between these two proteins could help treat these conditions by developing drugs that improve endothelial cell integrity in lymph vessels.

transported via collecting lymphatic vessels that are equipped with continuous zipper-like junctions preventing excessive leakage (*Baluk et al., 2007*; *Potente and Mäkinen, 2017*). The molecular mechanisms that establish and maintain such functionally specialised junctional features of different vessel types are poorly understood.

Vascular integrity is regulated by junctional adhesion molecules, of which the adherens junction molecule vascular endothelial (VE)-cadherin and the tight junction molecule Claudin 5 (CLDN5) have received particular attention. Dysregulation of VE-cadherin can result in junctional disruption, increased vessel permeability and severe (lymph)edema formation (*Frye et al., 2015*; *Hägerling et al., 2018*; *Yang et al., 2019*). Interestingly, different effects of gene disruption in mice have been observed depending on the organ and the vessel type. For example, endothelial loss of VE-cadherin in adult mice results in junctional disruption and increased vascular permeability in the heart and lung, but not the skin and brain (*Frye et al., 2015*). LEC-specific deletion of VE-cadherin in adult mice similarly leads to organ-specific disruption of endothelial junctions affecting the mesenteric collecting vessels but not dermal lymphatic vessels (*Hägerling et al., 2018*; *Yang et al., 2019*). Unlike VE-cadherin expressed in all ECs, CLDN5 is absent from certain phenotypically 'leaky' vessels (*Benz et al., 2019*). For instance, in the skin, CLDN5 is not expressed in blood vessels that are permissive to VEGF-induced vascular leakage (*Honkura et al., 2018*). In line with a function in junctional stabilization, CLDN5 is critical for the establishment and maintenance of the BBB (*Greene et al., 2018*; *Nitta et al., 2003*). CLDN5 is however highly expressed in the phenotypically 'permeable' lymphatic endothelium including lymphatic capillaries (*Baluk et al., 2007*), where its function is not known.

The junctional adhesion molecules are intracellularly associated to the actin cytoskeleton (*Dejana et al., 2009*). The molecular control of endothelial cytoskeletal dynamics presents therefore a second important leverage point that regulates vascular integrity. Quiescent endothelia are characterized by a balance of actin stabilization and myosin-based actin pulling forces that are constantly applied to endothelial junctions. Elevation of myosin-based actin contractility activates endothelial junctions. Rho GTPases are key regulators of such cytoskeletal dynamics (*Dorland and Huveneers, 2017*). Differential expression and function of Rho GTPase regulators, such as the guanosine

nucleotide exchange factor Vav3, has been proposed to contribute to barrier diversity across different blood vessels (*Hilfenhaus et al., 2018*). Upstream regulators of Rho GTPases may provide another level for organ- and vessel-type specific barrier regulation.

Here, we identified EphrinB2/EphB4 signalling as a critical and selective regulator of collecting lymphatic vessel integrity. Inducible EC-specific deletion of *Efnb2* or *Ephb4* disrupted lymphatic endothelial junctions in several organs while dermal and pulmonary blood vessel barrier function was not compromised. Studies in primary human LECs further showed that inhibition of EphrinB2/EphB4 signalling led to reduction in junctional CLDN5 while VE-cadherin was not affected. LEC-specific deletion of *Cldn5* in postnatal mice did not however fully recapitulate the junctional phenotype observed in *Efnb2/Ephb4*-deficient vessels. Interestingly, we found that basal EphrinB2/EphB4 signalling regulates junctional localisation of CLDN5 and junction stability via Rac1/Rho-mediated control of cytoskeletal contractility in primary LECs. Our results suggest that EphrinB2/EphB4 signalling provides a potential therapeutic target for intervention of diseases associated with abnormal lymphatic vessel permeability.

## Results

### Endothelial deletion of *Ephb4* or *Efnb2* disrupts cell-cell junctions selectively in lymphatic vessels of the skin

EphrinB2 and EphB4 play important roles in the embryonicand early postnatal development of blood and lymphatic vessels (*Adams et al., 1999*; *Gerety et al., 1999*; *Mäkinen et al., 2005*; *Zhang et al., 2015*). To study the role of EphrinB2/EphB4 signalling in the remodeling and quiescent vasculature, we conditionally deleted *Ephb4* or *Efnb2* in postnatal mice using the tamoxifen-inducible *Pdgfb-iCreER^T2^iresGFP* line (*Claxton et al., 2008*). We studied the effect on the dermal vasculature of the ear where *Pdgfb-CreER^T2^* targets the endothelium of all blood vessels and collecting lymphatic vessels, but not lymphatic capillaries (*Wang et al., 2017*). Gene deletion was induced by 4-Hydroxytamoxifen (4-OHT) administration at 3 weeks of age, when dermal endothelial cell proliferation has stopped (*Figure 1—figure supplement 1A*). Specificity of the *Pdgfb-CreER^T2^*-mediated recombination was confirmed using the *R26-mTmG* reporter in whole-mount stained ears (*Figure 1A* left panel).

Immunostaining of the ear skin of a 7-week-old 4-OHT-treated *Ephb4^flox/flox^;Pdgfb-CreER^T2^* mouse revealed abnormal collecting vessels with ectopic expression of the lymphatic capillary marker LYVE1 (*Figure 1A*) and disruption of VE-cadherin⁺ cell-cell junctions (*Figure 1B*). A similar, albeit more severe phenotype was observed in the *Efnb2^flox/flox^;Pdgfb-CreER^T2^* mice already 2 weeks after 4-OHT administration (*Figure 1C,D*). Unexpectedly, the cell-cell junctions of lymphatic capillaries not targeted by the *Pdgfb-CreER^T2^* transgene also appeared disorganized in the *Efnb2* mutant mice (*Figure 1—figure supplement 1B*), suggesting secondary effects caused by disruption of collecting vessels. Gross morphology of the blood vessels and the architecture of blood endothelial junctions were however unaltered in both mutants (*Figure 1A–D*). We next performed a modified Miles assay, to assess whether barrier function of the blood endothelium was affected by endothelial deletion of *Efnb2*. In agreement with a lack of morphological alterations of the junctions, permeability of the skin or lung vasculature was not changed in *Efnb2* mutants compared to control littermates under homeostasis (*Figure 1E*). To confirm these findings, we deleted endothelial *Efnb2* using another tamoxifen inducible Cre line, the *Cdh5-CreER^T2^* (*Wang et al., 2010*). Efficient depletion of EphrinB2 was shown exemplarily by Western blot from total lung lysates 8 days after the first tamoxifen administration (*Figure 1F*). However, no significant changes were observed either under homeostasis, or after VEGF- or histamine-induced vascular leakage in the lungs of *Efnb2^flox/flox^;Cdh5-CreER^T2^* mutant compared to Cre-negative littermate controls (*Figure 1F*).

In summary, these results demonstrate that loss of EphrinB2/EphB4 signalling in the postnatal vasculature leads to disruption of dermal lymphatic endothelial cell-cell junctions, while dermal and pulmonary blood endothelial cell junctions and barrier integrity are not altered.

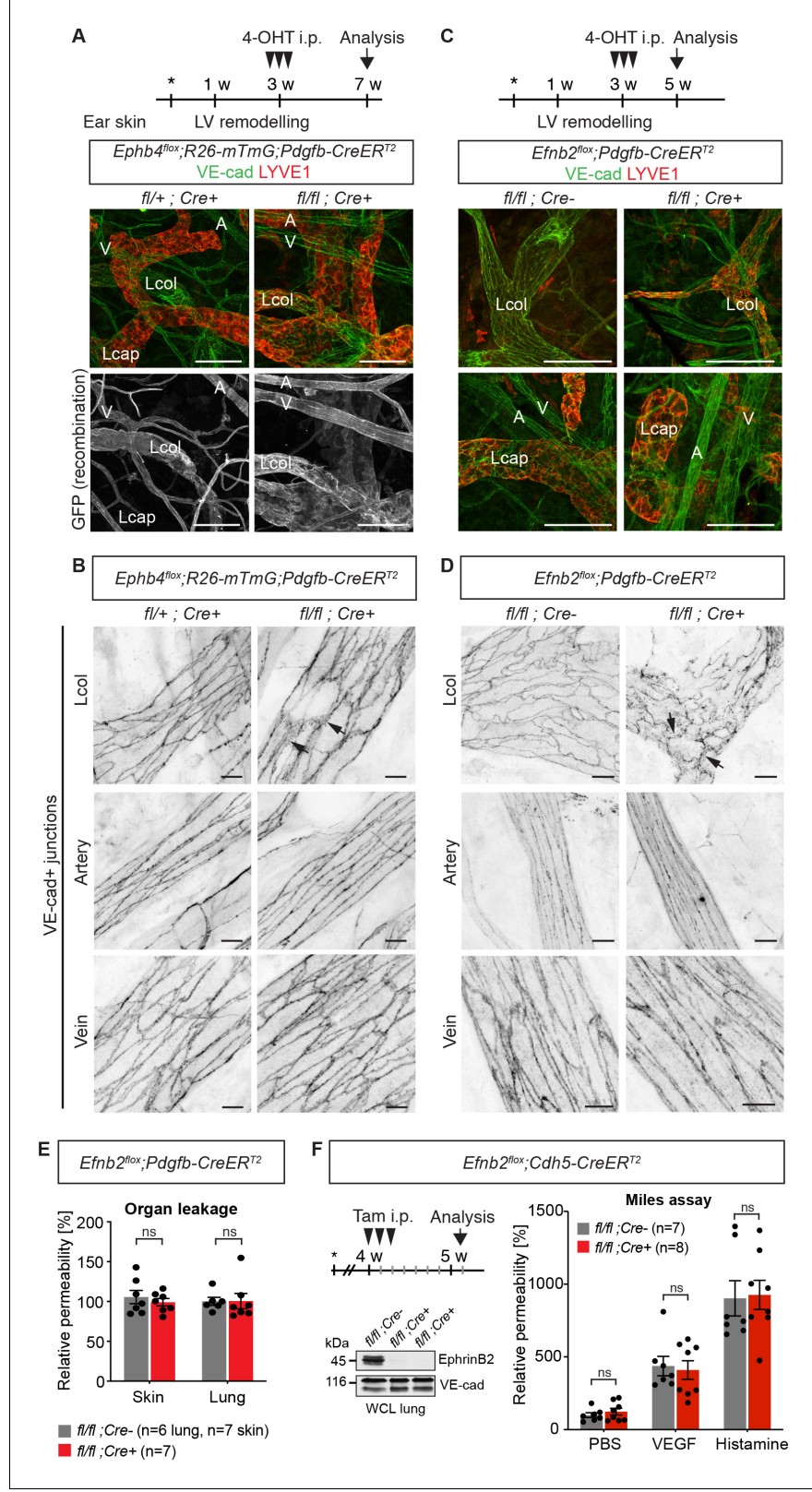

**Figure 1.** Endothelial deletion of *Ephb4* or *Efnb2* selectively disrupts dermal collecting lymphatic vessels. (**A**) Experimental scheme for *Ephb4* deletion in the mature vasculature by three consecutive intraperitoneal (i.p.) 4-OHT injections (arrowheads) into 3-week-old *Ephb4^flox^;R26-mTmG;Pdgfb-CreER^T2^* mice. Ear skin whole-mount immunofluorescence from 7-week-old *Ephb4* mice using antibodies against VE-cadherin (green) and LYVE1 (red)

*Figure 1 continued on next page*

*Figure 1 continued*

or GFP (single channel images). GFP expression demonstrates Cre recombination in arteries (A), veins (V) and LYVE1⁻ collecting lymphatic vessels (Lcol), but not LYVE1⁺ lymphatic capillaries (Lcap) in control ear. Mutant collecting vessels show abnormal LYVE1 expression. (B) Ear skin whole-mount immunofluorescence of 7-week-old *Ephb4* mice using an antibody against VE-cadherin. Note altered morphology of collecting lymphatic vessel junctions (arrow) in *Ephb4* mutant compared to heterozygous littermates. (C) Experimental scheme for *Efnb2* deletion in the mature vasculature by 3 consecutive 4-OHT injections (arrowheads) into 3-week-old *Efnb2^flox*; *Pdgfb-CreER^T2* mice. Ear skin whole-mount immunofluorescence of 5-week-old *Efnb2* mice using antibodies against VE-cadherin (green) and LYVE1 (red). (D) Ear skin whole-mount immunofluorescence of 5-week-old *Efnb2* mice using an antibody against VE-cadherin. Note altered morphology of collecting lymphatic vessel junctions (arrow) in *Efnb2* mutant compared to Cre negative littermates already after 2 weeks of Cre induction. (E) In vivo basal permeability assay in skin and lung of 5-week-old *Efnb2* mutants and Cre negative littermates. Data represent mean ± s.e.m. (n = 6–7 mice from two independent experiments). *Efnb2* deletion does not impact on basal barrier function of skin and lung blood vasculature. (F) Experimental scheme for *Efnb2* deletion using the *Cdh5-CreER^T2* line and three consecutive tamoxifen injections (arrowheads). Vascular leakage in the skin of 5-week-old *Efnb2* mutants and Cre negative littermates was induced with VEGF or histamine. Note, endothelial deletion of *Efnb2* does not impact on junctional regulation in leakage-induced dermal blood vasculature. Data represent mean ± s.e.m. (n = 7-8 mice from two independent experiments). Western blot from total lung lysates 8 days after the first tamoxifen administration showing depletion of EphrinB2 in Cre⁺ mice. VE-cadherin was used as a loading control. Source data for panels (E,F) are provided. Scale bars: 100 μm (A,C), 10 μm (B,D).
The online version of this article includes the following source data and figure supplement(s) for figure 1:

**Source data 1.** Flow cytometric analysis of endothelial cell proliferation in postnatal mouse ear skin.
**Figure supplement 1.** *Pdgfb-CreER^T2*-mediated deletion of *Efnb2* in mature collecting vessels leads to defective cell-cell junctions in lymphatic capillaries.
**Figure supplement 1—source data 1.** Measurement of blood vessel permeability in *Efnb2* mutants and control littermates.

## EphrinB2/EphB4 signalling regulates integrity of LEC junctions in different organs

To further investigate the role of EphrinB2/EphB4 signalling in the regulation of LEC junctions, we deleted *Efnb2* or *Ephb4* selectively in lymphatic endothelia using tamoxifen-inducible *Prox1-CreER^T2* mice (*Bazigou et al., 2011*). Gene deletion was induced by 4-OHT administration at postnatal day (P) 12, when remodeling of the ear vasculature into lymphatic capillaries and collecting vessels is initiated (*Lutter et al., 2012*) and increased mechanical tension at lymphatic junctions is expected (*Dorland and Huveneers, 2017*). GFP reporter expression demonstrated efficient and specific Cre-mediated targeting of the lymphatic vasculature (*Figure 2A*). *Efnb2* gene deletion was confirmed by qRT-PCR analysis of dermal LECs sorted from mutant and control ears (*Figure 2—figure supplement 1A*). Loss of *Efnb2* was also evidenced by defective lymphatic valve formation (*Figure 2A*), consistent with the previously described role of EphrinB2-EphB4 signalling in this process (*Mäkinen et al., 2005*; *Zhang et al., 2015*). Whole-mount staining for CLDN5 revealed disrupted cell-cell junctions and disintegration of the endothelial layer in *Efnb2*-deficient collecting lymphatic vessels (*Figure 2A*). Similar defects were observed after LEC-specific deletion of *Ephb4* (*Figure 2A*). Ultrastructural analysis of the *Efnb2* mutant ear vasculature using transmission electron microscopy confirmed disruption of cell-cell junctions, characterized by large intercellular gaps (*Figure 2B,C*) and convoluted junctions with increased junction overlap compared to controls (*Figure 2B,D*). We also observed signs of cell degeneration (*Figure 2B*). Deletion of *Efnb2* in the quiescent dermal lymphatic vasculature at 8 weeks and analysis at 18 weeks of age also showed junctional alterations, albeit less severe than in the juvenile mice (*Figure 2—figure supplement 1B*).

Next, we asked whether EphrinB2/EphB4 function in regulating LEC junctions is conserved in different vascular beds. Analysis of the outer lymphatic endothelial layer of the subcapsular sinus of inguinal lymph nodes that forms during embryonic development (*Bovay et al., 2018*) showed disorganized junctions in 4-week-old *Efnb2* and *Ephb4* mutant mice after deletion at P12 (*Figure 2E*). In addition, analysis of the mesenteric vasculature of P11 *Efnb2* and *Ephb4* mice neonatally (P4) treated with 4-OHT showed disorganisation of lymphatic endothelial cell-cell junctions in the mutant vessels compared to *Efnb2* heterozygous or Cre-negative controls (*Figure 2F*). As expected, defective lymphatic valves and chylothorax, indicative of valve dysfunction (*Nitschké et al., 2017*), were also

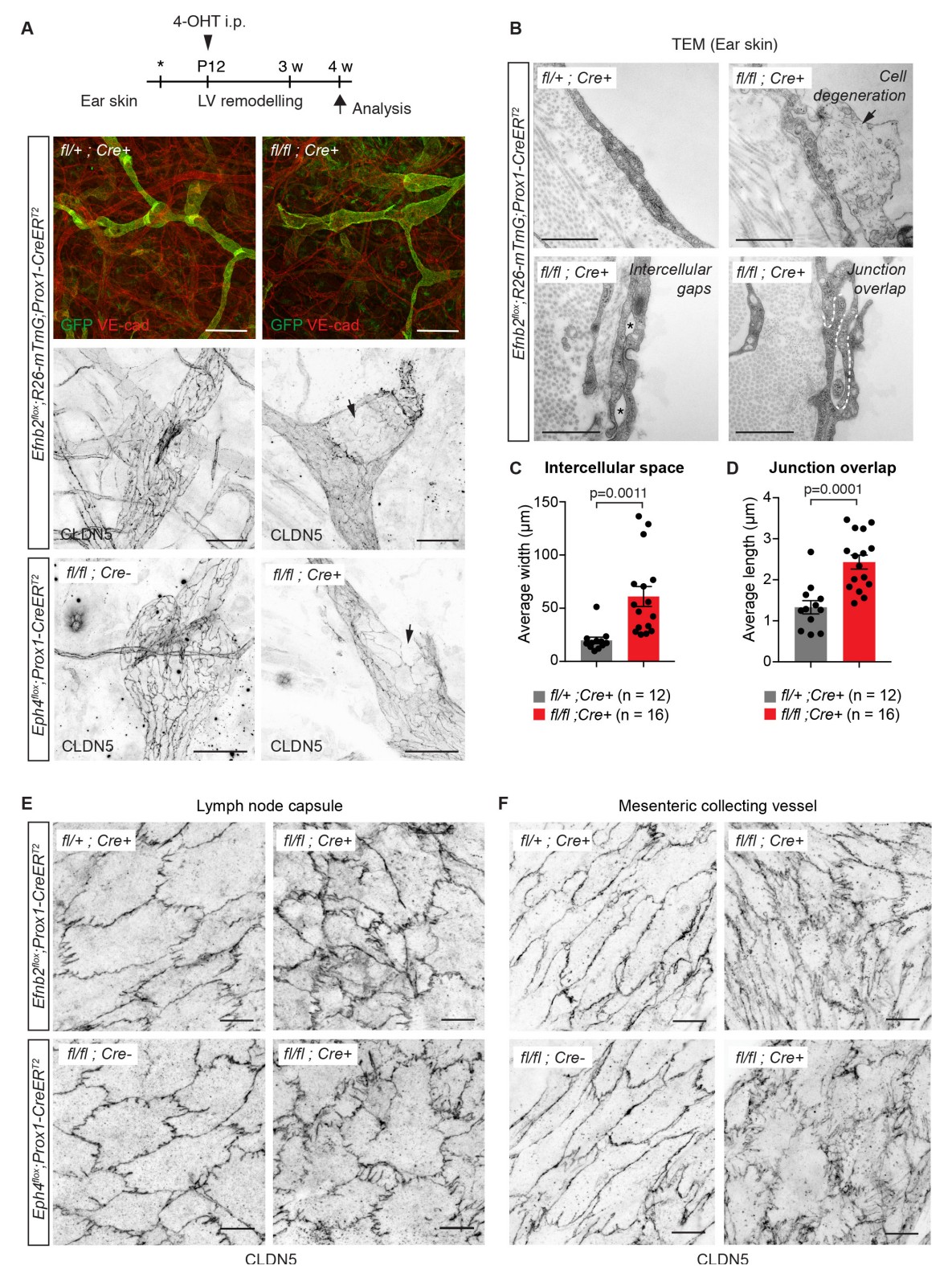

**Figure 2.** Lymphatic endothelial specific deletion of *Efnb2* or *Ephb4* disrupts endothelial cell-cell junctions in several organs. (**A**) Experimental scheme for LEC-specific deletion of *Efnb2* and *Ephb4* in the remodeling lymphatic vasculature (LV) using the *Prox1-CreER^T2* line and a single 4-OHT treatment at P12 (arrowhead). Ear skin whole-mount immunofluorescence of 4-week-old *Efnb2* and *Ephb4* mice using antibodies against VE-cadherin (red) and GFP (green) (shown for *Efnb2* mice only) or CLDN5 alone (single channel images). GFP expression demonstrates Cre recombination in lymphatic

*Figure 2 continued on next page*

*Figure 2 continued*

endothelia. Note disintegration of collecting lymphatic vessels in *Efnb2* and *Ephb4* mutant mice (arrows). (B) Ultrastructural analysis of the ear vasculature using transmission electron microscopy showing abnormal features of LEC junctions in *Efnb2* mutant mice compared to heterozygous littermate: intercellular gaps (asterisks), cell degeneration (arrow) and increased junction overlap (white dotted line). (C,D) Quantification of intercellular gaps and junctional overlap. Data represent mean ± s.e.m., p-value, Two-tailed unpaired Student's *t*-test (n = 12 vessels/194 junctions/3 mice (fl/+) or n = 16 vessels/250 junctions/2 mice (fl/fl)). (E) Whole-mount immunofluorescence of the outer surface of the subcapsular sinus (SCS) of the inguinal lymph node of 4-week-old *Efnb2* and *Ephb4* mice using an antibody against CLDN5. (F) Whole-mount immunofluorescence of P11 mesenteric collecting vessels in *Efnb2* and *Ephb4* mice using an antibody against CLDN5. Source data for panels (C,D) are provided. Scale bars: 500 μm (A), 100 μm (A single channel images), 1 μm (B), 10 μm (E,F).

The online version of this article includes the following source data and figure supplement(s) for figure 2:

**Source data 1.** Quantification of LEC junction parameters in the skin.
**Figure supplement 1.** *Efnb2* deletion in 4-week-old and 18-week-old mice.
**Figure supplement 1—source data 1.** qRT-PCR analysis of *Efnb2* expression in FACS sorted dermal LECs.
**Figure supplement 2.** Dysfunctional lymphatic valves and chylothorax in *Efnb2* and *Ephb4* mutant mice.
**Figure supplement 2—source data 1.** Incidence of chylothorax in *Ephb4* and *Efnb2* mutant mice.
**Figure supplement 3.** Abnormal morphology of LEC junctions in the inguinal lymph node capsule of *Efnb2* mutant mice.
**Figure supplement 3—source data 1.** Quantification of LEC junction morphology in the inguinal lymph node capsule.

observed (*Figure 2—figure supplement 2A,B*). Taken together, these data demonstrate that EphrinB2/EphB4 signalling regulates junctional integrity of lymphatic vessels of different vascular beds in juvenile and adult mice.

## Valve dysfunction does not cause disruption of LEC junctions in collecting vessels

Next, we asked if junctional disruption in the *Efnb2/Ephb4*-deficient lymphatic vessels is a cell-autonomous defect or caused secondarily due to valve disruption and consequent alterations in the transvalvular flow pattern affecting the vessel wall downstream of the valve (*Wilson et al., 2018*). Analysis of the P11 neonatal mesenteric vasculature of the *Efnb2^GFP* reporter mice showed that *Efnb2* expression was not restricted to valves but was present in all LECs of collecting vessels (*Figure 3A*). In addition, defects in cell-cell junctions were observed in *Efnb2*-deficient lymphatic vessel regions both upstream and downstream of the valve (*Figure 3B*). To quantify the state of the LEC junctions, we described four junctional categories based on previous definition of blood endothelial junctions (*Neto et al., 2018*): linear junctions, thick/reticular junctions, jagged junctions and discontinuous junctions as exemplified in *Figure 3C*. Linear morphology was previously described to represent mature stable junctions while reticular and jagged morphologies indicate active remodeling (*Dorland and Huveneers, 2017*). Finger-like structures were found in all lymphatic junctional categories, and were thus not included in the definition. In wild type vessels, linear junctions were most frequent in both the upstream and downstream regions of the valve (*Figure 3C*). Discontinuous junctions were not observed (*Figure 3C*), and were thus considered pathological. *Efnb2*-deficient vessels showed an almost complete loss of linear junctions at the expense of an increase in discontinuous junctions in collecting vessels, both upstream and downstream of the defective valves (*Figure 3C*). Quantification of the morphology of LEC junctions in the lymph node capsule (*Figure 2E*) confirmed the abnormal high prevalence of discontinuous junctions in the *Efnb2*-deficient mice (*Figure 2—figure supplement 3*).

To directly test if valve dysfunction caused by loss of *Efnb2* contributes to the disruption of LEC junctions in collecting vessels, we generated a tamoxifen-inducible Cre line that specifically targets lymphatic valves. To this end, a BAC-transgenic mouse line with expression of *CreER^T2* controlled by the regulatory elements of the lymphatic valve-specific *Cldn11* gene (*Takeda et al., 2019*) was generated (Ortsäter and Mäkinen, unpublished data). Specificity and efficiency of Cre-mediated recombination was validated using the *R26-mTmG* reporter (*Figure 3D* and *Figure 3—figure supplement 1A*). As expected, neonatal deletion of *Efnb2* using the *Cldn11-CreER^T2* mice resulted in malformed lymphatic valves, characterized by disorganisation of PROX1^high cells, in mesenteric lymphatic vessels (*Figure 3D* and *Figure 3—figure supplement 1B*). However, quantification of the state of the cell junctions in regions of collecting vessels located upstream or downstream of morphologically abnormal Cre-targeted (GFP^+) valve showed no alterations compared to the control (*Figure 3E*).

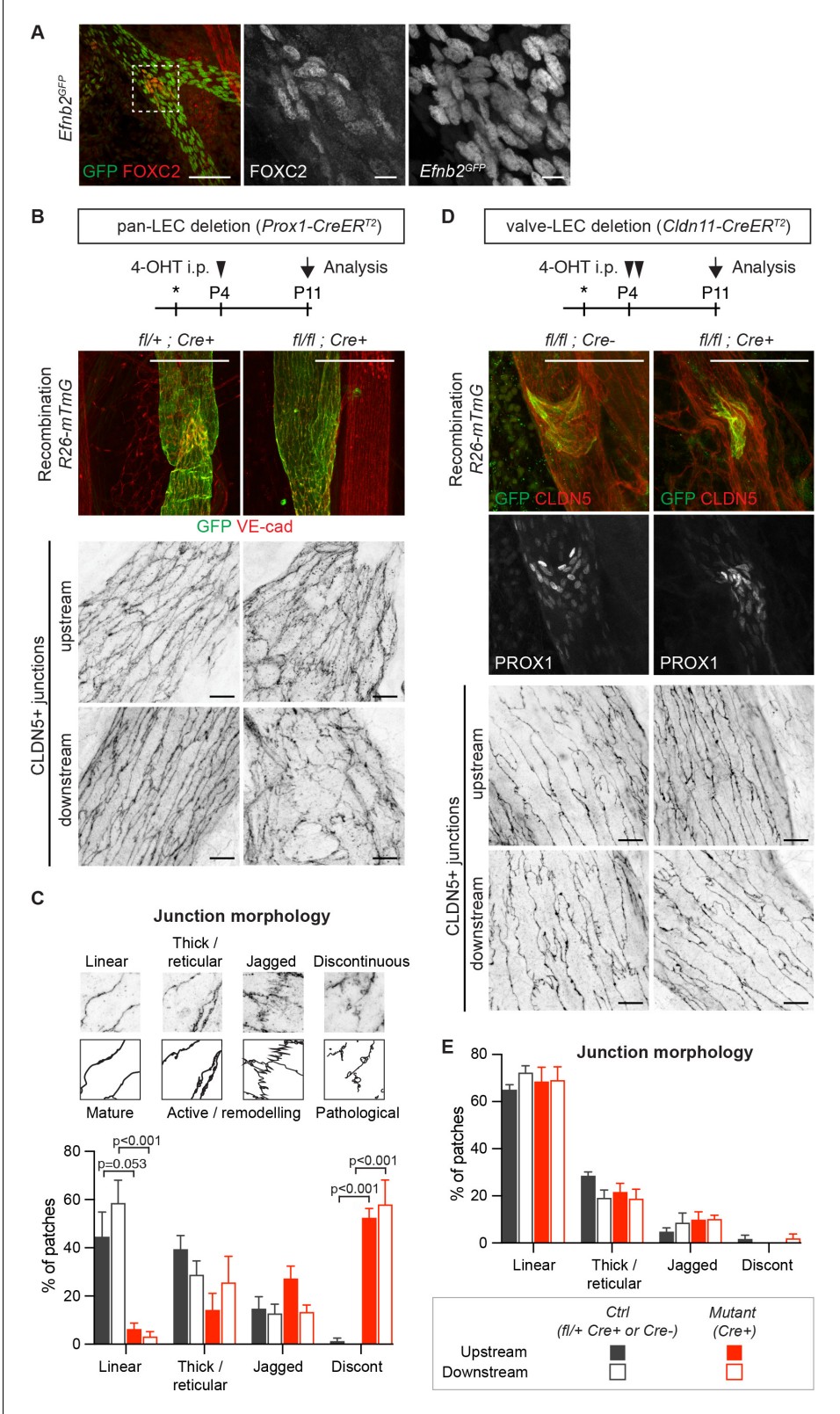

**Figure 3.** Lymphatic valve-specific deletion of *Efnb2* leads to valve defects but unaltered LEC junctions on the collecting vessel wall. (**A**) Whole-mount immunofluorescence of mesenteric lymphatic vessels of P11 *Efnb2*<sup>GFP</sup> mice using antibodies against GFP (green) and FOXC2 (red). Enlarged view of the boxed area with single channel images show valve-restricted FOXC2 expression but widespread GFP (reporting *Efnb2*) expression. (**B**) *Figure 3 continued on next page*

*Figure 3 continued*

Experimental scheme for neonatal *Efnb2* deletion in all LECs (pan-LEC) using the *Prox1-CreER^T2* mice. Top panels: Whole-mount immunofluorescence of P11 mesenteric vessels of *Efnb2^flox;R26-mTmG;Prox1-CreER^T2* mice using antibodies against VE-cadherin (red) and GFP (green). Note pan-LEC Cre deletion (GFP expression), and valve defect in the *Efnb2* mutant. Lower panels: Visualization of LEC junctions in vessels regions downstream or upstream of the valve using CLDN5 antibodies. (**C**) Quantification of junction morphology in CLDN5-stained mesenteric vessels upon pan-LEC *Efnb2* deletion. The defined junctional categories are illustrated above: linear (mature), thick/reticular and jagged (active), and discontinuous (pathological) junctions. Data represent mean ± s. e.m. adjusted p-value, Multiple *t*-tests, (n = 64–96 patches/3 mice per genotype and region). (**D**) Experimental scheme for neonatal *Efnb2* deletion in valve LECs using the *Cldn11-CreER^T2* mice. Top panels: Whole-mount immunofluorescence of P11 mesenteric vessels of *Efnb2^flox;R26-mTmG;Cldn11-CreER^T2* mice using antibodies against CLDN5 (red), GFP (green) and PROX1 (single channel images). Lower panels: Visualization of LEC junctions in vessels regions downstream or upstream of the valve using CLDN5 antibodies. Note valve-restricted Cre deletion (GFP expression), and misshaped valves but unaffected LEC junctions in the *Efnb2* mutant. (**E**) Quantification of junction morphology in CLDN5-stained mesenteric vessels upon valve-LEC specific *Efnb2* deletion. Data represent mean ± s.e.m., (n = 64 patches/2 mice per genotype and region). Source data for panels (**C,E**) are provided. Scale bars: 100 µm (**A, B, D** merge images), 10 µm (**A, B, D** single channel images).

The online version of this article includes the following source data and figure supplement(s) for figure 3:

**Source data 1.** Quantification of LEC junction morphology in the mesentery.
**Figure supplement 1.** Valve-specific deletion of *Efnb2* using the *Cldn11-CreER^T2* line.
**Figure supplement 1—source data 1.** Quantification of valve phenotype upon valve-specific deletion of *Efnb2*.

These results show that the disruption of lymphatic endothelial cell-cell junctions in *Efnb2*-deficient collecting vessels is not caused secondarily by valve dysfunction. Instead they suggest a collecting vessel LEC-autonomous function of EphrinB2/EphB4 signalling in regulating junctional integrity of lymphatic vessels.

## Selective reduction of junctional CLDN5 upon loss of basal EphrinB2/EphB4 signalling

Next, we studied the mechanism by which EphrinB2/EphB4 signalling regulates lymphatic endothelial junctions in primary human dermal lymphatic endothelial cells (HDLECs). To first test if the signalling pathway is activated in a basal state, EphB4 was immunoprecipitated using EphrinB2-Fc from HDLECs treated with control or *EFNB2* siRNA. Western blot analysis of the immunoprecipitations for phospho-tyrosine (4G10) showed basal phosphorylation of EphB4, which was abolished upon *EFNB2* silencing (*Figure 4A*). EphB4 protein levels on the surface of *EFNB2* silenced LECs were not however compromised as stimulation with crosslinked EphrinB2-Fc could potently activate EphB4 protein (*Figure 4A*).

To induce an acute loss of basal EphB4 signalling, we incubated HDLECs with an EphrinB2 function blocking antibody (B11), which binds to EphrinB2, thereby preventing ligation to and activation of the EphB4 receptor (*Abéngozar et al., 2012*). Staining for VE-cadherin revealed formation of thick/reticular and jagged HDLEC junctions, characterized by an increase in perpendicularly oriented VE-cadherin and junctional overlaps (*Figure 4B*). These morphological changes in junctions were observed already 3 hr after EphrinB2 inhibition (*Figure 4B*) and became more pronounced over time (*Figure 4—figure supplement 1*). To assess the functional consequence of EphrinB2 inhibition on monolayer permeability, confluent HDLECs grown on transwell filters were pre-incubated for 3 hr with the B11 antibody. Permeability to 40 kDa FITC-dextran was substantially increased in EphrinB2 inhibited compared to control HDLECs (*Figure 4C*). Notably, while the junctional localisation of VE-cadherin was minimally affected by EphrinB2 blockade, we observed reduction of junctional CLDN5 from VE-cadherin⁺ junctions (*Figure 4D*). A similar phenotype was observed after siRNA-mediated silencing of *EFNB2* (*Figure 4—figure supplement 2A,B*). Surprisingly, depletion of VE-cadherin in HDLECs did not result in disrupted CLDN5 junctions (*Figure 4E* and *Figure 4—figure supplement 3*), suggesting that the localisation of CLDN5 at LEC junctions is independent of VE-cadherin. This is in contrast to BECs where junctional expression of CLDN5 is dependent on VE-cadherin (*Taddei et al., 2008*).

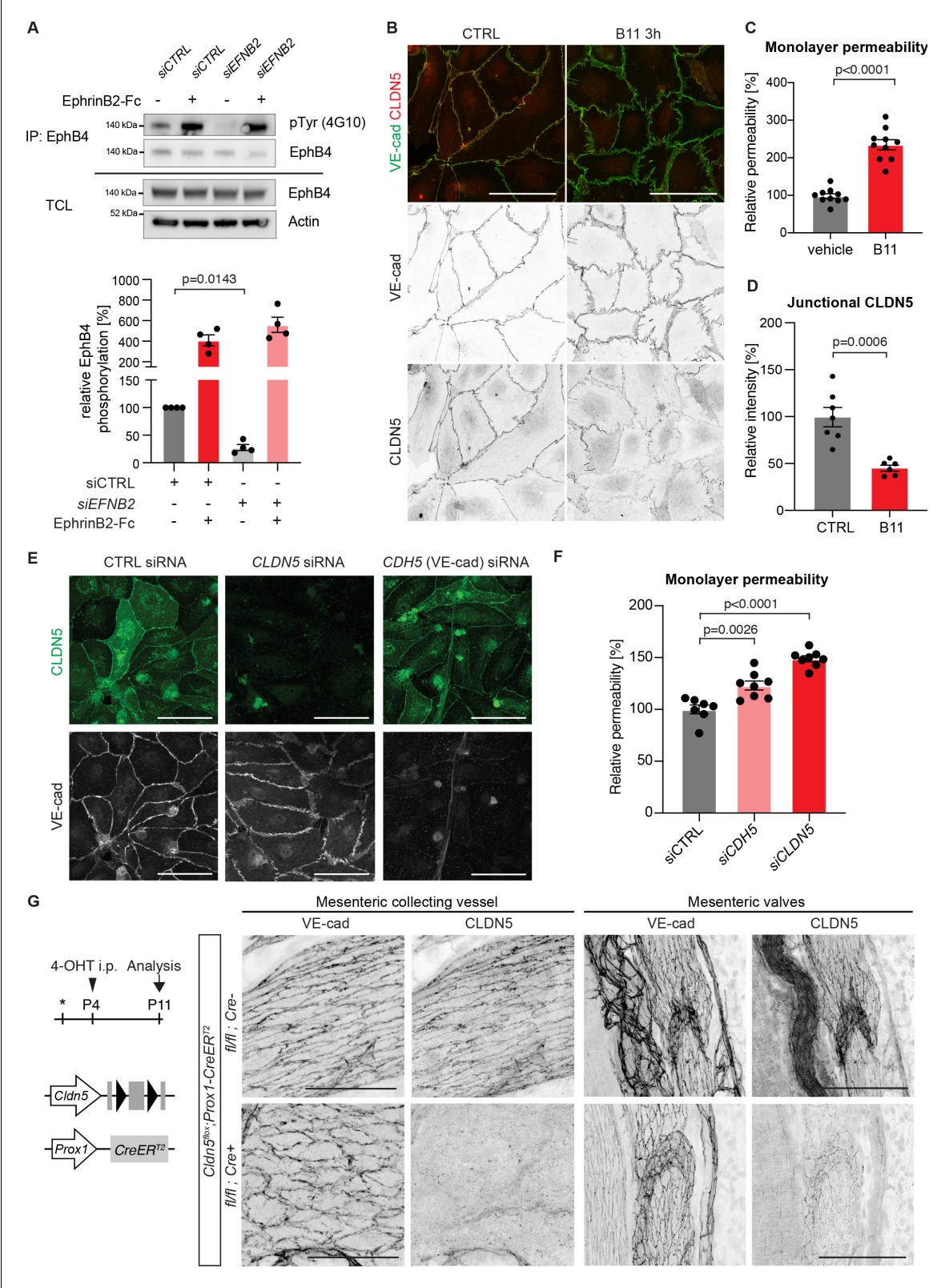

**Figure 4.** Loss of junctional CLDN5 occurs after EphrinB2 blockade in vitro but does not lead to breakdown of LEC junctions in vivo. (**A**) Top: Western blot of EphrinB2-Fc-precipitated EphB4 HDLECs for phosphotyrosine (4G10) and EphB4. Cells were treated with control (siCTRL) or EphrinB2 siRNA (*siEFNB2*), and stimulated with pre-clustered EphrinB2-Fc or control Fc. Total cell lysates (TCL) were immunoblotted for EphB4 and actin. Bottom: Quantification of relative EphB4 phosphorylation (adjusted to total precipitated EphB4 protein) from four independent experiments, showing EphrinB2-
*Figure 4 continued on next page*

Figure 4 continued

dependent basal EphB4 activity in unstimulated LECs. (B) Immunofluorescence of HDLECs treated with EphrinB2 blocking antibody (B11) for 3 hr using antibodies against VE-cadherin (green) and CLDN5 (red). Single channel images are depicted in grey. EphrinB2 blockade induces junction remodeling and removal of junctional CLDN5. (C) Quantification of HDLEC monolayer permeability to 40 kDa FITC-dextran showing increase upon B11 treatment for 3 hr (n = 10 replicates per condition from three independent experiments). (D) Quantification of junctional CLDN5 immunofluorescence in VE-cadherin⁺ junctions (n = 6–7 images from three independent experiments). (E) Immunofluorescence of HDLECs transfected with CTRL, *CLDN5* or *CDH5* (VE-cadherin) siRNA for 48 hr. Staining for CLDN5 (green) and VE-cadherin (grey) show disruption of VE-cadherin⁺ junctions upon *CLDN5* siRNA transfection. Note, CLDN5⁺ junctions are not affected upon VE-cadherin silencing. (F) Quantification of HDLEC monolayer permeability to 40 kDa FITC-dextran showing minor and moderate increase upon *CDH5* or *CLDN5* silencing, respectively (n = 7–8 replicates per condition from two independent experiments). (G) Whole-mount immunofluorescence of mesenteric collecting vessels and valves in P11 *Cldn5^flox^;Prox1-CreER^T2^* mice using antibodies against CLDN5 and VE-cadherin. Note efficient CLDN5 depletion apart from a few CLDN5 *hot spots* and altered morphology but no disintegration of the junctions. Data in A represent mean ± s.e.m. p value, One-sample *t*-test. Data in C, D, F represent mean ± s.e.m. p value, Two-tailed unpaired Student's *t*-test. Source data for panels (A,C,D,F) are provided. Scale bars: 100 μm (B, E, G mesenteric valves), 50 μm (G mesenteric collecting vessel). The online version of this article includes the following source data and figure supplement(s) for figure 4:

**Source data 1.** Quantification of the effects of EphrinB2 inhibition on EphB4 phosphorylation, endothelial monolayer permeability and junctional CLDN5.
**Figure supplement 1.** Loss of junctional CLDN5 after long-term EphrinB2 inhibition in primary LECs.
**Figure supplement 2.** The effect *EFNB2* silencing on CLDN5 and VE-cadherin.
**Figure supplement 3.** Silencing of *CDH5* and *CLDN5* in HDLECs.
**Figure supplement 4.** Generation of conditional *Cldn5* knock-out mice.

## Acute loss of CLDN5 is not sufficient to induce breakdown of LEC junctions

To test if depletion of junctional CLDN5 explained the increased monolayer permeability in EphrinB2-inhibited HDLECs, we first studied the effect of *CLDN5* silencing on the junctional organisation and monolayer permeability in HDLECs. Although *CDLN5* siRNA-treated LECs showed an increase in thick and reticular VE-cadherin⁺ junctions, monolayer permeability to 40 kDa FITC-dextran was only moderately increased (*Figure 4E,F*). Depletion of VE-cadherin by *CDH5* silencing also resulted in a minor increase in HDLEC monolayer permeability (*Figure 4F*).

To investigate the in vivo consequence of CLDN5 depletion on LEC junctions, we generated a floxed *Cldn5* allele (*Figure 4—figure supplement 4A,B*). Validation of the allele showed the expected loss of CLDN5 protein in *Cldn5^flox/flox^;PGK-Cre* (i.e. germline *Cldn5* homozygous) embryos (*Figure 4—figure supplement 4C*). Next, we conditionally deleted *Cldn5* in lymphatic endothelia at P4 using the tamoxifen-inducible *Prox1-CreER^T2^* mice (*Figure 4G*). Whole-mount immunostaining of mesenteries of P11 *Cldn5* mutant mice revealed efficient depletion of CLDN5 from lymphatic junctions, with the exception of CLDN5 *hot spots* remaining in particular in lymphatic valves (*Figure 4G*). *Cldn5* deficient vessels showed an increase in thick and reticular VE-cadherin junctions compared to control vessels (*Figure 4G*). However, we did not observe discontinuous junctions, characteristic of the lymphatic endothelium of EphrinB2/EphB4 mutant vessels. Loss of CLDN5 also did not result in obvious valve deformation or chylothorax formation at P11 (*Figure 4G* and data not shown).

Together, these results demonstrate that depletion of junctional adhesion molecule CLDN5 is not sufficient to explain the breakdown of lymphatic junctions observed upon EphrinB2 blockade in vitro or *Efnb2* deletion in vivo.

## Basal EphrinB2/EphB4 signalling controls the stability of LEC junctions via Rho GTPase-dependent regulation of the actin cytoskeleton

Endothelial junctions are tightly controlled via the actin cytoskeleton (*Dorland and Huveneers, 2017*). In human umbilical vein endothelial cells (HUVECs), blockade of EphrinB2/EphB4 signalling leads to increased F-actin stress fibre formation (*Abéngozar et al., 2012*). We thus sought to study if changes in the actin cytoskeleton upon loss of basal EphrinB2/EphB4 signalling contribute to disruption of HDLECs junctions. Like in HUVECs (*Dorland and Huveneers, 2017*), stable and quiescent HDLEC junctions are aligned by thick parallel cortical actin bundles. Remodeling HDLEC junctions are instead marked by a loss of cortical actin with concomitant increase in actin stress fibres and radial actin associated with perpendicular-oriented VE-cadherin+ junctions (*Sabine et al., 2015*). EphrinB2 inhibition with the B11 antibody induced disruption of cortical actin as shown by phalloidin

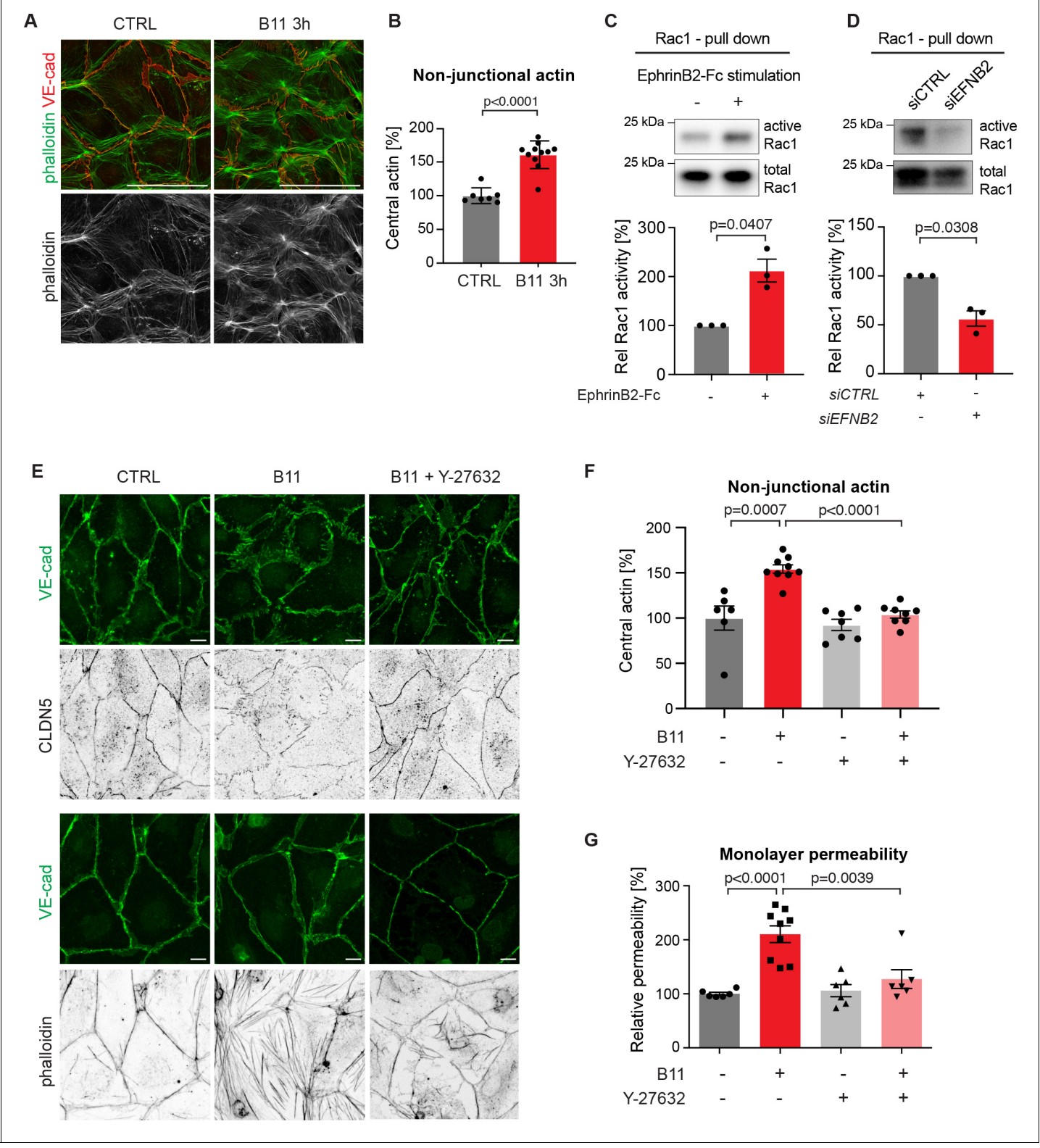

**Figure 5.** Basal EphrinB2/EphB4 signalling controls the stability of LEC junctions through regulation of Rho-mediated cytoskeletal contractility. Immunofluorescence of HDLECs for VE-cadherin (red) and phalloidin (green), showing increase in actin stress fibres after 3 h-treatment with the EphrinB2 blocking antibody (B11) compared to untreated controls (CTRL). Phalloidin single channel images are depicted in grey. (**B**) Quantification of relative central actin (total actin – junctional actin) from n = 7–11 images from three independent experiments shown in **A**. (**C, D**) Assessment of Rac1 activity in HDLECs after EphrinB2-Fc stimulation (30 min) (**C**) or siRNA-mediated *EFNB2* silencing (48 hr) (**D**) Rac1-GTP-pull down and total cell lysates

*Figure 5 continued on next page*

*Figure 5 continued*

were immunoblotted using Rac1 antibodies. Representative Western blots are shown. Quantification was done from three independent experiments. (**E**) Immunofluorescence of HDLECs incubated with EphrinB2 blocking antibody (B11) for 3 hr with and without pre-incubation of the ROCK inhibitor Y-27632 using antibodies against VE-cadherin (green) and CLDN5 (grey, upper panel) or phalloidin (grey, bottom panel). Note inhibition of EphrinB2 blockade (B11)-induced junctional and cytoskeletal effects by Y-27632 treatment. (**F**) Quantification of relative central actin (total actin – junctional actin) from n = 6–9 images from two independent experiments shown in **E**. (**G**) Quantification of HDLEC monolayer permeability to 40 kDa FITC-dextran. Note inhibition of EphrinB2 blockade (B11)-induced increase in permeability by Y-27632 treatment. (n = 6–9 replicates per condition from three independent experiments). Data in **B, F, G** represent mean ± s.e.m. p value, Two-tailed unpaired Student's *t*-test. Data in **C, D** represent mean ± s.e.m. p value, One-sample *t*-test. Source data for panels (**C-D**) are provided. Scale bars: 100 µm (**A**), 10 µm (**E**).

The online version of this article includes the following source data for figure 5:

**Source data 1.** Quantification of the effect of EphrinB2 blockade on the actin cytoskeleton and Rac1 activity.

staining of HDLECs (*Figure 5A*). Central actin (radial actin and actin stress fibres) was instead increased by 61% in B11-treated compared to untreated HDLECs (*Figure 5B*).

In the blood vasculature, the tyrosine receptor kinase Tie2 (TEK) signals through the Rho family GTPases Rac1 and RhoA to organize the cytoskeleton into a junction-fortifying arrangement that enhances barrier function (*Mammoto et al., 2007*). To test if Rac1 is similarly involved in EphrinB2/EphB4-mediated regulation of LEC junctions, we first assessed Rac1 activity in HDLECs after EphB4 activation or inactivation using a pull-down assay. Activation of EphB4 by crosslinked EphrinB2-Fc led to a 2-fold increase in Rac1 activity (*Figure 5C*). Conversely, inhibition of EphB4 activity by treatment with *EFNB2* siRNA led to a 44% decrease in Rac1 activity (*Figure 5D*).

Inhibition of Rac1 activity can lead to an increase in RhoA activity (*Wu et al., 2009b*). In blood ECs, spatio-temporal and antagonistic signalling of these Rho GTPases has been shown to regulate junctional integrity (*van Buul et al., 2014*). We thus tested whether interference with the Rho signalling pathway can inhibit junctional disruption caused by EphrinB2 blockade. To that end, we pre-incubated HDLECs with the well-established Rho kinase (ROCK) inhibitor Y-27632 for 45 min before treating the cells with the B11 antibody for 3 hr. Immunostaining revealed that junctional reduction of CLDN5 upon EphrinB2 blockade was inhibited in HDLECs pre-treated with the ROCK inhibitor Y-27632 (*Figure 5E*). Notably, cortical actin was maintained (*Figure 5E*) and the increase in central actin was markedly reduced in B11-treated HDLECs when ROCK activity was also inhibited (*Figure 5E,F*). ROCK inhibition alone did not affect the actin cytoskeleton (*Figure 5F*) or monolayer permeability of control HDLECs to 40 kDa FITC-dextran (*Figure 5G*). However, it efficiently inhibited EphrinB2 blockade-induced increase in monolayer permeability (*Figure 5G*).

We conclude that continuous basal EphrinB2/EphB4 signalling has a critical homeostatic function in regulating Rac1/Rho-mediated cytoskeletal organisation and thereby CLDN5-mediated lymphatic junction barrier integrity.

## Discussion

Endothelial barrier function is maintained via cell-cell junctions that need to be tightly controlled to allow selective permeability without compromising the overall junctional integrity. Here we identify the transmembrane ligand EphrinB2 and its tyrosine kinase receptor EphB4 as critical regulators of collecting lymphatic vessel integrity. We found that continuous basal EphrinB2/EphB4 signalling maintains cell junction stability selectively in LECs by controlling Rac1/Rho-mediated regulation of the actin cytoskeleton and CLDN5 localisation.

Although endothelial cell-cell junctions of all blood and lymphatic vessels share common basic structural features and molecular composition, there are important differences that reflect different vessel functions (*Potente and Mäkinen, 2017*). Fluid uptake function of lymphatic capillaries is known as a critical attribute of the lymphatic vasculature and mediated via specialized highly permeable button-like junctions (*Baluk et al., 2007*). The collecting lymphatic vessels are instead equipped with continuous zipper-like junctions, similar to those found in most blood vessels, that prevent excessive leakage and thus ensure efficient lymph transport (*Baluk et al., 2007*). Altered collecting lymphatic vessel permeability has been implicated in a variety of pathological conditions, highlighting the need to understand the underlying mechanisms. For example, collecting vessel permeability is increased upon infection with *Yersinia pseudotuberculosis* (*Fonseca et al., 2015*), which results in

development of fibrosis (*Ivanov et al., 2016*), increased inflammation and compromised immunity due to reduced flow of lymph and immune cell transport to the lymph nodes (*Kuan et al., 2015*). Excessive leakage of lymph, caused by dysfunctional lymphatic endothelial junctions in the regions of collecting vessel valves, may also contribute to primary (inherited) lymphedema (*Mahamud et al., 2019*; *Sabine et al., 2015*).

EphrinB2 and EphB4 are critical regulators of both blood and lymphatic vessel development (*Adams et al., 1999*: *Gerety et al., 1999*; *Mäkinen et al., 2005*; *Zhang et al., 2015*), and they show continuous expression in endothelial cells of both vessel types in adult tissues (*Luxán et al., 2019*; *Mäkinen et al., 2005*). Unexpectedly, we found that postnatal deletion of *Efnb2* or *Ephb4* in endothelial cells using the *Pdgfb-CreER^{T2}* or *Cdh5-CreER^{T2}* selectively disrupted the integrity of collecting lymphatic vessels in multiple organs, but did not compromise the barrier function of dermal and pulmonary blood vessels. The different requirements of BECs and LECs for junctional stability may be related to different dependencies on cell-cell *vs.* cell-matrix adhesions and substrate stiffness. In contrast to arteries and veins, collecting lymphatic vessels lack an elastic lamina, have a thinner basement membrane and less mural cell support (*Potente and Mäkinen, 2017*). Collecting vessel LECs may thus experience a less rigid extracellular matrix (ECM), which has been associated with an increase in cell-cell coupling in favour of cell-ECM adhesions that are strengthened on stiffer matrix (*Polio et al., 2019*). Although this hypothesis would need to be tested experimentally, it is interesting to note that *Efnb2* is upregulated in LECs grown on soft in comparison to stiff matrices (*Frye et al., 2018*). However, blood endothelial progenitors showed an increase in expression of the arterial marker EphrinB2 cultured on stiff (arterial) matrices (*Xue et al., 2017*), suggesting that upregulation of *Efnb2* on soft matrices is an LEC-specific phenomenon.

Although disruption of LEC junctions was observed in both *Efnb2* and *Ephb4* deficient vessels, the phenotype appeared stronger and developed more rapidly upon *Efnb2* deletion. It is possible that EphrinB2 reverse signalling plays a role and accounts for this difference. Our previous work showed that genetic deletion of the PDZ binding domain of EphrinB2 led to defects in developmental lymphatic vessel remodeling and collecting vessel maturation (*Mäkinen et al., 2005*). Although this suggested a role for EphrinB2 reverse signalling, it was later shown that the PDZ mutation affects not only the ligand-mediated reverse but also dampens receptor-mediated forward signalling (*Zhang et al., 2015*). Genetic deletion and antibody-mediated inhibition of EphB4 further established the forward signalling through the receptor as being the main Ephrin signalling pathway for collecting vessel and valve development (*Zhang et al., 2015*). The basal EphrinB2-dependent phosphorylation of EphB4 we observed in LECs supports, although does not prove this notion. It is also possible that the turnover of EphrinB2 and EphB4 proteins is different, such that faster depletion of EphrinB2 upon genetic deletion could explain the rapid development of a severe phenotype.

Although we did not observe blood vessel defects in the skin and lung of *Efnb2* or *Ephb4* deleted mice, a role for EphB4 in maintaining vascular integrity in the heart was recently demonstrated (*Luxán et al., 2019*). Endothelial-specific deletion of *Ephb4* in blood vessels in adult mice resulted in a hypertrophic cardiac phenotype that was attributed to a dysfunctional caveolae-mediated recycling of focal adhesion and junctional molecules specifically in coronary capillaries but not in skeletal muscle capillaries (*Luxán et al., 2019*). It has been suggested that organ-specific differences exist in the susceptibility to disruption of vascular integrity due to differences in the magnitude of mechanical stress they constantly have to resist (*Frye et al., 2015*; *Hägerling et al., 2018*; *Luxán et al., 2019*). It is therefore possible that junctional alterations occur at a later time-point in *Efnb2/Ephb4*-deficient dermal or pulmonary blood vessels and/or in context of specific pathological conditions. However, it is also interesting to note that LEC-specific deletion of *Efnb2* or *Ephb4* using the *Prox1-CreER^{T2}* mice led to global disruption of junctional integrity of collecting lymphatic vessels in all tissues analysed (skin, mesentery and lymph node), suggesting a general role of this pathway in LEC junction maintenance. It would be of interest to assess the effect of disruption of collecting vessel junctions on the ability of the vessels to take up and transport fluid, but also to study the long-term functional consequences in the context of pathology such as adult-onset obesity or inflammation.

ECs are subjected to fluid shear stress (FSS) due to the friction between the ECs and the blood or lymph. FSS has been extensively described to regulate endothelial junctions and lymphatic valve morphogenesis *Gordon et al., 2020*. In agreement with previous studies (*Mäkinen et al., 2005*; *Zhang et al., 2015*), we found that loss of EphrinB2-EphB4 signalling resulted in lymphatic valve defects. This raised the possibility that altered flow patterns contribute to disruption of EC junctions

on the collecting vessel wall. Interestingly, genetic inactivation of the mechanosensitive transcription factor *Foxc2* also results in degeneration of lymphatic valves and disruption of LEC junctions (*Sabine et al., 2015*). The junctional phenotype in the *Foxc2*-deficient vessels is however restricted to the valve sinus, where flow recirculation maintains high FOXC2 expression (*Sabine et al., 2015*). Expression of *Efnb2* in all LECs of collecting vessels instead suggested a role beyond valve areas. In agreement with this, LEC-specific deletion of *Efnb2* using the *Prox1-CreER^{T2}* resulted in junctional disruption in collecting vessels both upstream and downstream of the valve. In further support of a cell-autonomous function of EphrinB2 in all collecting vessel LECs, valve-specific deletion of *Efnb2* using the *Cldn11-CreER^{T2}* resulted in valve disruption but cell-cell junctions of the lymphatic vessel wall were not affected. Lymphatic capillaries also express *Efnb2* (*Bazigou et al., 2011*). However, we were not able to assess the potential cell-autonomous role of EphrinB2 in the formation and mainte-nance of button junctions in lymphatic capillaries because they were affected secondary to deletion of *Efnb2* in collecting vessels. Interestingly, LEC-specific deletion of VE-cadherin was previously shown to result in junctional defects in and fragmentation of lacteal lymphatic capillaries in the intes-tinal villi secondary to defects in the mesenteric collecting lymphatic vessels (*Hägerling et al., 2018*). These observations suggests that normal function of collecting vessels, ensuring efficient lymph flow, is critical for the maintenance of LEC junctions in lymphatic capillaries.

ECs are interconnected via VE-cadherin-mediated adherens junctions that have important roles in the development and maintenance of lymphatic vessels (*Dartsch et al., 2014*; *Hägerling et al., 2018*; *Yang et al., 2019*). Interestingly, however, organ-specific differences exist in the role of VE-cadherin in vessel maintenance. LEC-specific deletion of *Cdh5* (encoding VE-cadherin) in adult mice resulted in deterioration of mesenteric collecting vessels, while dermal lymphatic vessels were not affected (*Hägerling et al., 2018*). This is different from the phenotype in both vascular beds observed upon loss of EphrinB2/EphB4 signalling, which suggests the involvement of VE-cadherin-independent mechanisms of junction disruption. The role of other molecular players was further sup-ported by lack of effect of EphrinB2/EphB4 inhibition on VE-cadherin localisation or levels in HDLECs in vitro. Instead, we found that EphrinB2/EphB4 blockade resulted in reduction in the junctional localisation of CLDN5. Genetic deletion of *Cldn5* in LECs in vivo did not however fully recapitulate the phenotype observed in *Efnb2/Ephb4*-deficient lymphatic vessels. Although the morphology of cell junctions was altered, we could not observe a complete breakdown of the junctions as in the *Efnb2/Ephb4* mutants. Although our data suggest a more prominent role of CLDN5 in controlling LEC junction integrity compared to VE-cadherin, cross-talk and compensation between CLDN5 and VE-cadherin is likely to occur.

Quiescent endothelia are characterized by a balance of actin stabilization and myosin-based actin pulling forces that are constantly applied to endothelial junctions. Elevation of myosin-based actin contractility activates endothelial junctions, but abnormally increased contractility can ultimately lead to adherens junction disruption and loss of vascular integrity (*Faurobert et al., 2013*). In agreement with previous findings in HUVECs (*Abéngozar et al., 2012*), we found that loss of EphrinB2/EphB4 signalling in primary LECs led to an increase in actin stress fibres that was attributed to increased Rho activity. Although in vivo validation remains to be done, our findings support a concept that continuous basal activation of EphrinB2/EphB4 signalling provides a homeostatic control of Rac1/Rho-mediated cytoskeletal contractility to regulate LEC junction integrity. Interestingly, loss of (valve-specific) FOXC2 and FOXC1 in LECs was recently shown to result in increased actin stress fibre formation and Rho activity, suggesting similar mechanisms of junctional regulation in valve LECs (*Norden et al., 2020*). It may therefore not be surprising that ablation of the adhesion mole-cule CLDN5 was not sufficient to induce a breakdown of LEC junctions in the absence of activation of the cytoskeleton. A similar concept has been suggested in established blood vessels, whereby basal Tie2 receptor signalling mediates stabilization of pulmonary endothelial junctions via the actin cytoskeleton even in the absence of VE-cadherin (*Frye et al., 2015*).

In conclusion, we show that continuous basal activation of EphrinB2/EphB4 signalling provides a critical homeostatic mechanism regulating Rac1/Rho-mediated cytoskeletal contractility in LECs and integrity of cell junctions in collecting lymphatic vessels in vivo. Activation of the EphB4 receptor or downstream signalling components may thus provide an opportunity to overcome pathological hyperpermeability and restore basal permeability of collecting lymphatic vessels, thereby presenting a potential strategy for selective modulation of lymphatic vessel function.

# Materials and methods

## Key resources table

| Reagent type (species) or resource | Designation | Source or reference | Identifiers | Additional information |
|---|---|---|---|---|
| Genetic reagent (*Mus musculus*) | *Efnb2*$^{flox}$ | *Grunwald et al., 2004* | *Efnb2*$^{tm4Kln}$; RRDI: MGI:2182626 | |
| Genetic reagent (*Mus musculus*) | *Ephb4*$^{flox}$ | *Martin-Almedina et al., 2016* | | |
| Genetic reagent (*Mus musculus*) | *Pdgfb-iCreER*$^{T2}$*iresGFP* | *Claxton et al., 2008* | *Tg(Pdgfb-icre/ERT2)*$^{1Frut}$; RRDI: MGI:3793852 | |
| Genetic reagent (*Mus musculus*) | *R26-mTmG* | *Muzumdar et al., 2007* | *Gt(ROSA)26Sor*$^{tm4(ACTB-tdTomato,-EGFP)Luo}$; RRDI: MGI:3716464 | |
| Genetic reagent (*Mus musculus*) | *Prox1-CreER*$^{T2}$ | *Bazigou et al., 2011* | *Tg(Prox1-cre/ERT2)*$^{1Tmak}$; RRDI: MGI:5617984 | |
| Genetic reagent (*Mus musculus*) | *Efnb2*$^{GFP}$ | *Davy and Soriano, 2007* | *Efnb2*$^{tm2Sor}$; RRDI: MGI:3526818 | |
| Genetic reagent (*Mus musculus*) | *Cldn11-CreER*$^{T2}$ | This paper | | H Ortsäter and T Mäkinen, manuscript in preparation. |
| Genetic reagent (*Mus musculus*) | *Cldn5*$^{flox}$ | This paper | *Cldn5*$^{tm1a(EUCOMM)Wtsi}$; MGI:5473167 | Obtained from The European Conditional Mouse Mutagenesis Program (EUCOMM). |
| Cell line (*Homo sapiens*) | Dermal lymphatic endothelial cell (normal, juvenile, male) | PromoCell | Cat# C12216 | Primary cell line isolated from foreskin, tested negative for mycoplasma contamination |
| Antibody | anti-EphrinB2 (Human single chain variable fragment) | *Abéngozar et al., 2012* | B11 | (80 µg/ml) |
| Antibody | anti-human IgG (Goat polyclonal) | Jackson ImmunoResearch | Cat# 109-005-098 | (5 µg/ml) |
| Antibody | anti-human/ mouse/rat EphrinB2 (Goat polyclonal) | R and D Systems | Cat# AF496 | (0.5 µg/ml) |
| Antibody | anti-human EphB4 (Goat polyclonal) | R and D Systems | Cat# AF3038 | (0.5 µg/ml) |
| Antibody | anti-human VE-cadherin (Goat polyclonal) | Santa Cruz Biotechnology | C19, Cat# sc-6458 | (IF: 0.2 µg/ml; WB 2 µg/ml) |
| Antibody | anti-human/ mouse CLDN5 (Rabbit polyclonal) | Invitrogen | Cat# 34–1600 | (0.5 µg/ml) |

*Continued on next page*

*Continued*

| Reagent type (species) or resource | Designation | Source or reference | Identifiers | Additional information |
|---|---|---|---|---|
| Antibody | anti-phosphotyrosine 4G10 (Mouse monoclonal) | Merck | Cat# 05–321 | (0.5 µg/ml) |
| Antibody | anti-β-actin (Rabbit polyclonal) | Cell Signalling Technologies | Cat# 4967 | (0.1 µg/ml) |
| Antibody | anti-mouse Podoplanin (Syrian hamster monoclonal) | eBioscience | eBio8.1.1, PE; Cat# 12-5381-82; RRDI:AB_1907439 | (2 µg/ml) |
| Antibody | anti-mouse CD31 (Rat monoclonal) | eBioscience | 390, PE-Cy7, Cat# A14715; RRDI:AB_2534231 | (0.7 µg/ml) |
| Antibody | anti-mouse CD45 (Rat monoclonal) | eBioscience | 30-F11, PerCP-Cyanine5.5; Cat# 5-0451-82, RRDI:AB_1107002 or eFluor450; Cat# 48-0451-82; RRDI:AB_1518806 | (4 µg/ml) |
| Antibody | anti-mouse CD11b (Rat monoclonal) | eBioscience | M1/70, PerCP-Cyanine5.5; Cat# 45-0112-82; RRDI:AB_953558 or eFluor450; Cat# 48-0112-82; RRDI:AB_1582236 | (4 µg/ml) |
| Antibody | anti-TER-119 (Rat monoclonal) | eBioscience | TER-119, eFluor450; Cat# 48-5921-82; RRDI:AB_1518808 | (4 µg/ml) |
| Antibody | anti-Ki-67 (Rat monoclonal) | eBioscience | SolA15, eFluor 660; Cat# 50-5698-82; RRDI:AB_2574235 | (1:100) |
| Antibody | anti-human VE-cadherin (Mouse monoclonal) | Santa Cruz Biotechnology | F-8, Cat# sc-9989 | (2 µg/ml) |
| Antibody | anti-mouse LYVE1 (Rat monoclonal) | R and D Systems | Cat# MAB2125 | (1 µg/ml) |
| Antibody | anti-GFP (Rabbit polyclonal) | Abcam | Cat# ab290 | (1 µg/ml) |
| Antibody | anti-mouse FoxC2 (Sheep polyclonal) | R and D Systems | Cat# AF6989 | (2 µg/ml) |
| Sequence-based reagent | TaqMan probe: *Gapdh* | ThermoFisher Scientific | Cat# Mm_03302249_g1 | |
| Sequence-based reagent | TaqMan probe: *Efnb2* | ThermoFisher Scientific | Cat# Mm00438670_m1 | |
| Sequence-based reagent | siRNA: human *CLDN5* | Dharmacon | Cat# D-011409-03-0010 | |
| Sequence-based reagent | siRNA: human *CDH5* | Dharmacon | Cat# D-003641-03-0010 | |
| Sequence-based reagent | siRNA: human *EFNB2* | Dharmacon | Cat# L-003659-00-0005 | |
| Sequence-based reagent | siRNA: negative control, human | Qiagen | Cat# 1027281 | |

*Continued on next page*

*Continued*

| Reagent type (species) or resource | Designation | Source or reference | Identifiers | Additional information |
|---|---|---|---|---|
| Peptide, recombinant protein | Human EphrinB2-Fc | R and D Systems | Cat# 7397-EB | (0.5 µg/ml: activation assay, 3 µg: IP) |
| Chemical compound, drug | ROCK inhibitor (Y-27632) | Sigma | Cat# SCM075 | (10 µM) |
| Commercial assay or kit | Rac1-GTP pulldown assay | Cytoskeleton Inc | Cat# BK030 | |

## Mice

*Efnb2^{flox}* (**Grunwald et al., 2004**), *Ephb4^{flox}* (**Martin-Almedina et al., 2016**), *Pdgfb-iCreER^{T2}iresGFP* (**Claxton et al., 2008**), *R26-mTmG* (**Muzumdar et al., 2007**), *Prox1-CreER^{T2}* (**Bazigou et al., 2011**) and *Efnb2^{GFP}* (**Davy and Soriano, 2007**) mice were described previously. *Ephb4* and *Efnb2* mutant mice were analysed and showed the same phenotype both in the presence and absence of the *R26-mTmG* reporter allele. *Cldn11-CreER^{T2}* were generated using BAC transgenesis, by inserting a cDNA encoding *CreER^{T2}* under the control of the regulatory elements of the lymphatic valve-specific *Cldn11* gene (H Ortsäter and T Mäkinen, manuscript in preparation).

ES cell line containing 'knockout-first' *Cldn5* allele (*Cldn5^{tm1a(EUCOMM)Wtsi}*) was obtained from The European Conditional Mouse Mutagenesis Program (EUCOMM). After obtaining germ line transmission, *LacZ-neo* cassette was removed by crossing with a FlpO deleter strain (**Wu et al., 2009a**) followed by mating to C57BL/6J for at least three generations.

For induction of Cre-mediated recombination, 4-hydroxytamoxifen (4-OHT, H7904, Sigma Aldrich) was administered by intraperitoneal injection at P4 (1 × 50 µg, dissolved in Ethanol), or at P12 (1 × 1 mg) or P21-22 (3 × 1 mg, dissolved in peanut oil). For induction of Cre-mediated recombination at 4 and 8 weeks, tamoxifen (1–2 mg, T5648, Sigma Aldrich) was dissolved in peanut oil and administered by intraperitoneal injections on 3–5 consecutive days. All strains were maintained and analysed on a C57BL/6J background (backcrossed minimum six times). Experimental procedures were approved by the Uppsala Animal Experiment Ethics Board, Sweden (permit numbers: C416/12 and C130/15) and by the Landesamt für Natur, Umwelt und Verbraucherschutz (LANUV), Nordrhein-Westfalen, Germany (permit number: 8.87–50.10.36.09.063). Animals were kept in a barrier facility under pathogen–free conditions.

## Cell culture

Human primary dermal lymphatic endothelial cells (HDLEC from juvenile foreskin, cat. C12216) were obtained from PromoCell and tested negative for mycoplasma contamination. Cells were cultured on bovine Fibronectin-coated dishes in complete ECGMV2 medium (PromoCell) at 37° with 5% $CO_2$.

## RNA interference

Dharmacon siGENOME siRNA was used to knock-down human *CLDN5* (Cat# D-011409-03-0010) and *CDH5* (Cat# D-003641-03-0010). Dharmacon ON-TARGETplus SMARTpool siRNA was used to knock-down human *EFNB2* (Cat# L-003659-00-0005). As a control AllStars Negative Ctrl siRNA was used (Cat# 1027281, Qiagen). Cells were subjected to transfection 24 hr after plating, using Lipofectamine 2000 (Invitrogen) according to the manufacturer's instructions.

## In vitro permeability assay

To determine paracellular permeability, $4 \times 10^4$ HDLECs were seeded on fibronectin–coated Transwell filters (Costar 3413, 0.4 µm pore size; Corning) and grown to confluence. For EphrinbB2 blockade, HDLECs were serum-starved (EGMV2 basal medium + 0.5% FBS) and then incubated with EphrinB2 blocking antibody B11 (80 µg/ml) or isotype ctrl or PBS for 2 hr. To study the impact of Rho inhibition, HDLECs were pre-incubated with the ROCK inhibitor (Y-27632) for 30 min before the EphrinB2 blocking antibody B11 was added. After 2 hr, 0.25 mg/ml FITC-dextran (40 kDa; Sigma-

Aldrich) was added to the upper chamber of the transwells and transcellular diffusion was allowed for 1 hr. Fluorescence in the lower chamber was measured with a microplate reader (Synergy HTX Multi-Mode Microplate Reader), and monolayer integrity was confirmed by immunofluorescence staining for VE-cadherin after each assay.

## EphB4 activation

30 min prior to activation of EphB4 by its ligand EphrinB2, recombinant human EphrinB2-Fc (0.5 µg/ml, 7397-EB, R and D Systems) was clustered using Goat Anti-Human IgG (5 µg/ml, 109-005-098, Jackson ImmunoResearch).

## Immunoprecipitation and immunoblotting

For cell lysis and detection of phospho-tyrosine after immunoprecipitation, HDLECs were lysed in lysis buffer containing 20 mM Tris-HCl, pH 7.4, 150 mM NaCl, 2 mM $CaCl_2$, 1 mM $Na_3VO_4$, 1% Triton X-100, 0.04% $NaN_3$, and 1 × complete EDTA-free protease inhibitors (Roche). EphB4 was immunoprecipitated from cell lysates by incubation with 3 µg human EphrinB2-Fc (7397-EB, R and D Systems) and 30 µl Protein G-Sepharose for 3 hr at 4°C. Immunocomplexes were washed five times with lysis buffer and analysed by SDS-PAGE.

Mouse lungs were homogenized with an Ultra Turrax (IKA-Werke) in RIPA buffer containing 1% NP-40, 1% sodium deoxycholate, 0.01 M NaPi, 150 mM NaCl, 2 mM EDTA, 1 mM $Na_3VO_4$, and 2 × Complete EDTA-free protease inhibitors, followed by incubation for 4 hr at 4°C. Lysates were centrifuged at 4°C for 1 hr at 20,000 g and supernatant was used for direct blot analysis.

Total cell or organ lysates or immunoprecipitated material was separated by SDS-PAGE and transferred to PVDF membranes (88520, Thermo Fisher Scientific) by wet blotting. For detection of phosphorylated tyrosine, milk powder in the blocking buffer was replaced by 2% BSA, and 200 µM $Na_3VO_4$ was added. The following antibodies were used: Goat anti-EphrinB2 (AF496, R and D Systems, 0.5 µg/ml), goat anti-EphB4 (AF3038, R and D Systems, 0.5 µg/ml), goat anti-VE-cadherin (sc-6458, Santa Cruz Biotechnology, 0.2 µg/ml), rabbit anti-CLDN5 (34–1600, Invitrogen, 0.5 µg/ml), mouse anti-phosphotyrosine 4G10 (05–321, Merck, 0.5 µg/ml), rabbit anti-β-actin (4967, Cell Signalling Technologies, 0.1 µg/ml).

## Rac1 pull-down assay

Serum-starved (EGMV2 basal medium + 0.5% FBS) confluent HDLEC cultures remained resting or were stimulated for 30 min with 0.5 µg ml$^{-1}$ clustered recombinant human EphrinB2-Fc (7397-EB, R and D Systems) and subsequently subjected to Rac1-GTP pulldown assay according the manufacturer's instructions (BK030, Cytoskeleton Inc). Briefly, equal amounts of protein lysates were incubated for 1 hr at 4°C under gentle agitation with 10 µg PAK-RBD beads. After extensive washes PAK-RBD-bound proteins were denatured by the addition of 2 × Laemmli buffer and incubation at 97°C for 3 min. Samples were immunoblotted onto PVDF membranes (88520, Thermo Fisher Scientific), probed with mouse anti-Rac1 antibodies (ARC03, Cytoskeleton Inc) and corresponding donkey anti-mouse-HRP-conjugated secondary antibodies (715-035-151, Jackson ImmunoResearch). Membranes were visualized using a ChemiDoc MP imaging system (Biorad).

## Flow cytometry

Sorting of blood and lymphatic endothelial cells was done using ear skin of 4-weeks-old *Efnb2$^{flox}$; R26-mTmG;Prox1-CreER$^{T2}$* mice. Tissue were collected and immediately digested at 37°C in PBS supplemented with Collagenase IV (Life technologies) 10 mg/ml, DNase1 (Roche) 0.1 mg/ml and FBS 0.5% (Life technologies) for 25 min with vigorous shaking every 5 min. Collagenase activity was quenched by dilution with FACS buffer (PBS, 0.5% FBS, 2 mM EDTA) and digestion products were filtered twice through 70 µm nylon filters (BD Biosciences). Cells were again washed with FACS buffer and then processed for enrichment of CD31/PECAM1 positive cells using magnetic beads according to the manufactures instructions (Miltenyi). After enrichment, Fc receptor binding was blocked with rat anti-mouse CD16/CD32, (eBioscience). Samples were thereafter stained with anti-podoplanin (clone eBio8.1.1, conjugation eFluor660, 2 µg/ml), anti-CD31/PECAM1 (390, PE-Cy7, 0.7 µg/ml), anti-CD45 (30-F11, eFluor450, 4 µg/ml), anti-CD11b (M1/70, eFluor450, 4 µg/ml) and anti-TER-119 (TER-119, eFluor450, 4 µg/ml) all obtained from eBioscience. Prior to sorting cells were

incubated with Sytox blue (Life technologies) to label dead cells. Cells were sorted directly to tubes containing RNA extraction buffer (Qiagen) on a BD FACSAria III cell sorter configured with four lasers (405, 488, 561 and 643 nm) and equipped with a 85 µm nozzle. Single cells were gated from FSC-A/SSC-A, FSC-H/FSC-W and SSC-H/SSC-W plots followed by exclusion of all cells with a eFluor450/Sytox blue positive signal. For compensation, the AbC anti-rat/hamster compensation bead kit (Life Technologies) was used. ECs (CD31/PECAM1$^+$) were then sorted as either LECs (PDPN$^+$GFP$^+$) or BECs (PDPN$^-$GFP$^-$Tomato$^+$).

Tissue samples for FACS analysis of proliferating cells were taken from wild-type P3 pups (developing ear buds) and from P10, P21 or 5 weeks old mice (ear skin). Tissue samples were dissected, cut into small pieces and digested in Collagenase IV (Life technologies) 4–10 mg/ml, DNase1 (Roche) 0.1–0.2 mg/ml and FBS 1–10% (Life technologies) in PBS at 37°C for 10 to 30 min (depending on stage) with vigorous shaking every 5 min. Samples were processed as described above except that no enrichment for CD31/PECAM1 positive cells was performed. Staining of cell surface markers was performed by incubation with antibodies targeting podoplanin (eBio8.1.1, PE), CD31/PECAM1 (390, PE-Cy7), CD45 (30-F11, PerCP-Cyanine5.5), CD11b (M1/70, PerCP-Cyanine5.5) all obtained from eBioscience. After staining cells were thoroughly washed with PBS and stained for dead cells using the blue LIVE/DEAD fixable dead cell stain kit from Life technologies. Thereafter cells were fixed and permeabilized using the Foxp3/Transcription factor staining kit (eBioscience) according to the manufacturer's instructions. Finally cells were incubated with rat serum and then a Ki-67 antibody (SolA15, eFluor 660, eBioscience, 1:100). Cells were analysed on a BD LSR Fortessa cell analyzer configured with five lasers (355, 405, 488, 561 and 643 nm). Compensation was performed using the anti-rat/hamster compensation bead kit and the ArC amine reactive compensation bead kit (Life technologies). Single viable cells were gated as described above and in *Martinez-Corral et al., 2020* and dead cells were excluded in the 355-UV laser dump channel. FMO controls were used to set up the subsequent gating scheme to obtain cell populations and quantification of proliferating cells. Flow data were processed using FlowJo software version 10.5.0 (TreeStar).

## qRT PCR analysis

Ear skin was dissected and minced in 5 mg/ml Collagenase II (Roche) and 0.2 mg/ml DNaseI (Roche) digestion mix. From FACS-sorted LECs (CD31$^+$/PDPN$^+$) total RNA was extracted by RNeasy Micro kit (QIAGEN) and all obtained RNA was reverse transcribed using oligo dT (SuperScript III First-Strand Synthesis System, Invitrogen). cDNA was pre-amplified using the TaqMan PreAmp Master Mix Kit. Gene expression levels were analysed using TaqMan Gene Expression Assay (AppliedBiosystems) and the StepOne Plus Real-Time PCR system (Applied Biosystems) following manufacturer's instructions. Relative gene expression levels were normalized to GAPDH. The following probes were used: *Gapdh* Mm_03302249_g1, *Efnb2* Mm00438670_m1 (ThermoFisher Scientific).

## In vivo permeability assay in the skin

A modified Miles assay for the induction of vascular permeability in the skin was performed as described previously (*Frye et al., 2015*). Evans blue dye (Sigma-Aldrich) was injected into the tail vein (100 µl of a 1% solution in PBS) of *Efnb2* mutant and control mice (n = 2–3 animals/group from three independent experiments). After 15 min, 50 µl PBS, 100 ng murine VEGF$_{165}$ in 50 µl PBS, or 225 ng histamine in 50 µl PBS was injected intradermally into the shaved back skin. 30 min later, skin areas were excised and extracted with formamide for 5 d at RT, and the concentration of the dye was measured at 620 nm with a spectrophotometer (Shimadzu).

## In vivo basal permeability assay of the skin and lung

*Efnb2* mutant and control mice (n = 3–4 animals/group from two independent experiments) were intravenously injected with Evans blue dye (Sigma Aldrich, 100 µl of a 1% solution in PBS), 30 min later sacrificed, and the body circulation was perfused with PBS. Weight-adjusted skin patches and lungs were removed and extracted with formamide for 5 d at RT. The concentration of the dye was measured at 620 nm with a spectrophotometer (Shimadzu).

## Transmission electron microscopy (TEM)

Specimens were fixed in 1% glutaraldehyde and 4% formaldehyde in 0.1 M phosphate buffer, pH 7.4, followed by post-fixation in 1% osmium tetroxide, dehydration in acetone and embedding in Epon LX112 (Ladd Research Industries). 150 nm sections were stained with toluidine blue to select regions of interest. 80 nm sections were cut with a Leica Ultracut UCT microtome and imaged using Tecnai Spirit transmission electron microscope (Fei Europe) and Quemesa CCD camera (Olympus Soft Imaging Solutions GMBH). Measurements were performed from TEM micrographs at a 11000 × magnification using ImageJ.

## Immunofluorescence

For whole-mount immunostaining, tissues were fixed in 4% paraformaldehyde (PFA, all expect CLDN5) or ice-cold methanol (CLDN5 immunostaining) overnight at 4°C, permeabilised in 0.3% Triton-X100 in PBS (PBSTx) and blocked in PBSTx plus 3% BSA (blocking buffer). Primary antibodies were incubated at 4°C overnight in blocking buffer. After washing in PBSTx, the samples were incubated with Alexa Fluor-conjugated secondary antibodies in blocking buffer, before further washing and mounting in Dako Fluorescence Mounting Medium. For staining of HDLECs, cells were fixed with 4% PFA in PBS for 20 min at RT or with ice-cold methanol (CLDN5 immunostaining) for 20 min at 4° and permeabilized using 0.5% Triton-X100 in PBS for 5 min at RT followed by blocking with 3% BSA in PBSTx for 1 hr. Primary antibodies were incubated for 1 hr at RT, washed twice with PBSTx and subsequently incubated with secondary antibodies for 45 min at RT before further washing and mounting in Dako Fluorescence Mounting Medium.

The following antibodies were used: goat anti-VE-cadherin (C19, cat. sc-6458, Santa Cruz Biotechnology, 2 μg/ml), mouse anti-VE-cadherin (F8, cat. sc-9989, Santa Cruz Biotechnology, 2 μg/ml), rat anti-LYVE1 (cat. MAB2125, R and D Systems, 1 μg/ml), rabbit anti-GFP (ab290, Abcam, 1 μg/ml), rabbit anti-CLDN5 (34–1600, Invitrogen, 0.5 μg/ml), sheep anti-mouse FoxC2 (AF6989, R and D Systems, 2 μg/ml). Secondary antibodies conjugated to AF488, AF647, Cy2, Cy3 or Cy5 were obtained from Jackson ImmunoResearch (all used 1:200). Additionally, Alexa Fluor 568 Phalloidin (A12380, ThermoFisher Scientific) and Alexa Fluor 594 Phalloidin (A12381, ThermoFisher Scientific) were used.

## Image acquisition

Confocal images of tissues (whole-mount immunostaining) and cells represent maximum intensity projections of Z-stacks that were acquired using Leica SP8 inverted microscope with HCX PL APO CS 10 ×/0.40 DRY or HC PL APO CS2 63 ×/1.30 GLYC objectives and Leica LAS-X software.

## Image quantification

To quantify the state of in vivo LEC junctions (*Figure 3C,E*), four junctional categories were defined: straight, thick/reticular, jagged and discontinuous junctions. 2–3 images (66 μm$^2$) per mouse and region (upstream and downstream of the valve) were acquired from 2 (*Cldn11-CreER$^{T2}$*) - 3 (*Prox1-CreER$^{T2}$*) independent experiments and divided each image in 16 patches. Classification of each patch into the four categories was done manually and blinded. Wild type LECs of the inguinal lymph node capsule showed a more irregular junctional morphology (*Figure 2E* and *Figure 2—figure supplement 3*), therefore we defined three junctional categories: linear/thick/reticular, jagged and discontinuous junctions. Six images (80 μm$^2$) per genotype were acquired from two independent experiments and each image was divided into 25 patches. Classification of each patch into the three categories was done manually. To quantify valve morphology in the *Efnb2$^{flox}$;Cldn11-CreER$^{T2}$* mice, we analysed the organization and alignment of PROX1$^{high}$ nuclei within the valve region of Cre-targeted (GFP positive) P11 *Efnb2$^{flox}$;R26-mTmG;Cldn11-CreER$^{T2}$* mesenteries. We defined two categories: normal alignment of PROX1$^{high}$ nuclei and disorganized alignment of PROX1$^{high}$ nuclei along valve leaflets as illustrated in *Figure 3—figure supplement 1*. Classification of valves was done manually with 10–11 valves per mesentery from three littermate mice.

All other quantifications were done using Fiji ImageJ. For quantification of intercellular gaps and junctional overlap from TEM micrographs, 12–16 lymphatic vessels with 6 to 28 junctions per vessel were measured. The total number of junctions was 194 from three heterozygous control mice and 250 junctions from two *Efnb2* mutant mice. For quantification of central actin (*Figure 5B,F*), we applied a threshold for junctional VE-cadherin staining, created a threshold-based mask and pixel

intensities of the junctional actin fraction and overall actin pixel intensity was subtracted. 6–11 images (250 µm$^2$, maximum intensity projection images with 12 z-stacks) were acquired from 2 to 3 independent experiments. For junctional CLDN5 immunostaining pixel intensity measurements, we applied a defined threshold for junctional VE-cadherin staining, created a threshold-based mask and analysed pixel intensities of CLDN5 immunostaining in the junctions. CLDN5 pixel intensities (integrated density) were detected from 6 to 7 images (250 µm$^2$, maximum intensity projection images with 12 z-stacks) from three independent experiments. Detailed number of experiments are indicated in figure legends. Western blot signal quantifications were done using BioRad Image Lab Software.

## Statistics

GraphPad Prism was used for graphic representation and statistical analysis of the data. We used 2-tailed unpaired Student's t-test to compare between two means, assuming equal variance, multiple t-tests (with correction for multiple comparison using the Holm-Sidak method) to compare between multiple conditions and One-sample t-test to compare sample mean with a normalized control value = 1 or 100. Differences were considered statistically significant when p or adjusted $p < 0.05$.

## Acknowledgements

We thank Ralf Adams (Max Planck Institute for Molecular Biomedicine, Münster) for the *Cdh5-CreER$^{T2}$* mice, Marcus Fruttiger (University College London, London) for the *Pdgfb-iCreER$^{T2}$iresGFP* mice and Rüdiger Klein (Max Planck Institute for Neurobiology, Martinsried) for the *Efnb2$^{flox}$* mice. We also thank the BioVis facility (Uppsala University, Sweden) for flow cytometer usage and support, and Sofie Sjöberg and Sofie Lunell-Sergerqvist for technical assistance. Euro-BioImaging ERIC and Biocenter Oulu Electron Microscopy core facility supported by University of Oulu and Biocenter Finland are acknowledged for ultrastructural analysis.

This work was supported by the Swedish Cancer Society (CAN 2013/387), Knut and Alice Wallenberg Foundation (2015.0030 and 2018.0218), the European Research Council (ERC-2014-CoG-646849) and the Swedish Research Council (542-2014-3535) to TM. MF was supported by a postdoctoral fellowship from Lymphatic Education & Research Network and the European Union's Horizon 2020 research and innovation programme under the Marie Sklodowska-Curie grant agreement No 840189. SS was supported by a research fellowship from the Deutsche Forschungsgemeinschaft (STR 1538/1–1) and a non-stipendiary long-term fellowship from the European Molecular Biology Organization (ALTF 86–2017). JLM-T was supported by the Regional Government of Madrid (Angiobodies Programme BIPEDD2 S2011/BMD-2312) and the European Social Fund. LE was supported by the Academy of Finland (310986).

## Additional information

### Competing interests

John Wiseman: is affiliated with AstraZeneca. The author has no financial interests to declare. The other authors declare that no competing interests exist.

### Funding

| Funder | Grant reference number | Author |
| --- | --- | --- |
| Cancerfonden | CAN 2013/387 | Taija Mäkinen |
| Knut och Alice Wallenbergs Stiftelse | 2015.0030 | Taija Mäkinen |
| Knut och Alice Wallenbergs Stiftelse | 2018.0218 | Taija Mäkinen |
| Vetenskapsrådet | 542-2014-3535 | Taija Mäkinen |
| H2020 European Research Council | ERC-2014-CoG-646849 | Taija Mäkinen |

| | | |
|---|---|---|
| Lymphatic Education & Research Network | | Maike Frye |
| Horizon 2020 Framework Programme | 840189 | Maike Frye |
| Deutsche Forschungsgemeinschaft | STR 1538/1-1 | Simon Stritt |
| European Molecular Biology Organization | ALTF 86-2017 | Simon Stritt |
| Regional Government of Madrid | BIPEDD2 S2011/BMD-2312 | Jorge L Martínez-Torrecuadrada |
| European Social Fund | | Jorge L Martínez-Torrecuadrada |
| Suomen Akatemia | 310986 | Lauri Eklund |

The funders had no role in study design, data collection and interpretation, or the decision to submit the work for publication.

## Author contributions
Maike Frye, Conceptualization, Formal analysis, Investigation, Methodology, Writing - original draft, Project administration, Writing - review and editing; Simon Stritt, Henrik Ortsäter, Mika Kaakinen, Formal analysis, Investigation, Writing - review and editing; Magda Hernandez Vasquez, Andres Vicente, Investigation, Writing - review and editing; John Wiseman, Lauri Eklund, Jorge L Martínez-Torrecuadrada, Resources, Writing - review and editing; Dietmar Vestweber, Conceptualization, Supervision, Writing - review and editing; Taija Mäkinen, Conceptualization, Formal analysis, Supervision, Funding acquisition, Writing - original draft, Project administration, Writing - review and editing

## Author ORCIDs
Maike Frye (iD) https://orcid.org/0000-0002-6257-7636
Simon Stritt (iD) http://orcid.org/0000-0002-4299-4934
Andres Vicente (iD) https://orcid.org/0000-0003-0718-6816
Lauri Eklund (iD) http://orcid.org/0000-0002-3177-7504
Jorge L Martínez-Torrecuadrada (iD) https://orcid.org/0000-0002-8240-6623
Dietmar Vestweber (iD) http://orcid.org/0000-0002-3517-732X
Taija Mäkinen (iD) https://orcid.org/0000-0002-9338-1257

## Ethics
Animal experimentation: Experimental procedures were approved by the Uppsala Animal Experiment Ethics Board, Sweden (permit numbers: C416/12 and C130/15) and by the Landesamt für Natur, Umwelt und Verbraucherschutz (LANUV), Nordrhein-Westfalen, Germany (permit number: 8.87-50.10.36.09.063).

## Decision letter and Author response
Decision letter https://doi.org/10.7554/eLife.57732.sa1
Author response https://doi.org/10.7554/eLife.57732.sa2

# Additional files
## Supplementary files
• Transparent reporting form

## Data availability
All relevant data generated or analysed during this study are included in the manuscript and supporting files. Source data files have been provided for Figures 1-5, Figure 1—figure supplement 1, Figure 2—figure supplements 1-3 and Figure 3—figure supplement 1.

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
