## [Decision Letter]

**Acceptance summary:**

This work examines the effects of genetic manipulations on the collecting tubules of the lymphatics in vivo, with selected culture experiments to complement the in vivo data. The lymphatic vasculature exhibits heterogeneity that has been largely overlooked, but likely is important for understanding the impact of therapeutics, so the focus on collecting tubules is timely. This work uses elegant genetic tools and readouts to examine how the Ephrinb2-EphB4 signaling axis affects different types of lymphatic endothelial cell-cell junctions, and places this signaling axis upstream of Rho-Rac signaling that affects the cytoskeleton.

**Decision letter after peer review:**

Thank you for submitting your article "EphrinB2-EphB4 signalling provides Rho-mediated homeostatic control of lymphatic endothelial cell junction integrity" for consideration by *eLife*. Your article has been reviewed by three peer reviewers, including Victoria L Bautch as the Reviewing Editor and Reviewer #1, and the evaluation has been overseen by Jonathan Cooper as the Senior Editor.

The reviewers have discussed the reviews with one another and the Reviewing Editor has drafted this decision to help you prepare a revised submission.

As the editors have judged that your manuscript is of interest but additional experiments are requested before it is published, we would like to draw your attention to changes in our revision policy that we have made in response to COVID-19 (https://elifesciences.org/articles/57162). First, because many researchers have temporarily lost access to the labs, we will give authors as much time as they need to submit revised manuscripts. We are also offering, if you choose, to post the manuscript to bioRxiv (if it is not already there) along with this decision letter and a formal designation that the manuscript is "in revision at *eLife*". Please let us know if you would like to pursue this option. (If your work is more suitable for medRxiv, you will need to post the preprint yourself, as the mechanisms for us to do so are still in development.)

Summary:

In a study that is overall elegant and rigorous, Frye et al. report novel and interesting effects of the EphrinB2-EphB4 ligand-receptor signaling pathway in maintaining the integrity of collecting lymphatic vessel junctions. The authors had previously revealed that this pathway controls lymphatic valve formation. Here they extend these studies to show that the pathway maintains the integrity of collecting vessel junctions independently of the effects on valves. in vitro experiments complement those findings and provide insight into the molecular mechanisms that are at play when EphrinB2 and EphB4 signaling is prevented. The data reveal a striking decrease in Claudin5 in endothelial junctions, due to altered cytoskeletal adaptations, but not involving the canonical endothelial adhesion receptor VE-cadherin. The manuscript covers a timely topic and is well written. Overall the experiments have been well controlled and represent interesting new findings that are of importance for the vascular biology field, since leaky collecting lymphatic vessel junctions have been implicated in inflammatory diseases and obesity, and our understanding of the regulation of these junctions is still rudimentary.

Essential revisions:

1) Functional data for aberrant LEC junctions in vivo: The in vivo effects of manipulating *Efnb2-Ephb4* signaling on lymphatics relies primarily on histological analysis. Would it be possible to complement the data by functional dye uptake studies, using fluorescent dextran dye injections into the ear skin? One would expect that dye uptake should be unaffected because the *Pdgfb* Cre driver does not delete in lymphatic capillaries, but leaky collectors could be visualized this way. Comparison between the junctional phenotypes of *Cdh5-CreER^T2^* and *Pdgfb-CreER^T2^* mediated *Efnb2* and *Ephb4* deletions could also be informative, because the former does delete in lymphatic capillaries, hence this would tell if the pathway is selective for the collector junctions. In the case that further in vivo analysis of the phenotype cannot be accomplished, we request text revisions to acknowledge this caveat.

2) Evidence that Rac1/Rho signaling is involved in vivo: The authors observe a reduction of Claudin5 expression in LEC junctions after blocking EphrinB2 function, but Claudin5 knockout did not mimic the junctional defects seen in *Efnb2* or *Ephb4* mutants. Hence, additional mechanisms must be at play. The authors examine Rac1/Rho-mediated cytoskeletal organization. The cell culture work is nicely done and shows that Rock inhibition can rescue junctional and permeability defects induced by EphrinB2 function blocking antibodies. However, whether this accounts for the in vivo phenotype is not shown. Can pharmacological Rock inhibition rescue the mutant collecting junction phenotype? It should be possible to directly inject and determine if the leaky collecting vessels form improved zipper junctions. In the case that this further analysis cannot be accomplished, we request text revisions to acknowledge this caveat.

3) Antibody validation and comparison: have the authors validated the specificity and/or selectivity of the B11 function blocking antibody? For instance, through a comparison of junctional changes in siCTRL and siEFNB2 transfected cells, or phosphorylation of the EphB4 receptor, to make sure that the observed endothelial changes in response to B11 are indeed due to altered EphrinB2-EphB4 signaling. The authors state in the text (subsection “Selective reduction of junctional CLDN5 upon loss of basal EphrinB2/EphB4 signalling”, last paragraph) that the junctional defects with antibody incubation become more pronounced over time, but do not show the data. How do the LECs and the junctions look like after EphrinB2 and EphB4 siRNA knock down?

---

## [Author Response]

Essential revisions:1) Functional data for aberrant LEC junctions in vivo: The in vivo effects of manipulating Efnb2-Ephb4 signaling on lymphatics relies primarily on histological analysis. Would it be possible to complement the data by functional dye uptake studies, using fluorescent dextran dye injections into the ear skin? One would expect that dye uptake should be unaffected because the Pdgfb Cre driver does not delete in lymphatic capillaries, but leaky collectors could be visualized this way. Comparison between the junctional phenotypes of Cdh5-CreER^T2^ and Pdgfb-CreER^T2^ mediated Efnb2 and Ephb4 deletions could also be informative, because the former does delete in lymphatic capillaries, hence this would tell if the pathway is selective for the collector junctions. In the case that further in vivo analysis of the phenotype cannot be accomplished, we request text revisions to acknowledge this caveat.

We agree with the reviewers that functional data would be valuable. Dye uptake studies in the ear skin cannot distinguish between dye absorption (capillaries) and transport (collecting vessels) defects, but, as suggested by the reviewers, collecting vessel-specific deletion driven by the *Pdgfb-CreER^T2^* transgene should circumvent this problem. Unexpectedly, however, we found that LEC junctions in lymphatic capillaries were also affected in the *Efnb2^flox/flox^;Pdgfb-CreER^T2^* mice (new Figure 1—figure supplement 1B). We did not study this phenotype in detail since it developed secondary to disruption of collecting vessels. Interestingly, a similar observation was made in the gut vasculature upon LEC-specific deletion of VE-cadherin that led to junctional changes in and fragmentation of lacteal lymphatic capillaries in the intestinal villa secondary to defects in the mesenteric collecting lymphatic vessels (Hagerling et al., 2018). This is now discussed in the Discussion section.

We also considered assays on dissected collecting vessels as done by Prof. Michael Davis (School of Medicine, University of Missouri, US) or Dr. Joshua Scallan (Department of Molecular Pharmacology and Physiology, University of South Florida, US), which would be valuable in assessing dye leakage from the collecting vessels. At this point it would however take a significant time to organise the collaboration. To acknowledge the lack of functional assessment of dye uptake/transport, we have added the following sentence in the text:

“It would be of interest to assess the effect of disruption of collecting vessel junctions on the ability of the vessels to take up and transport fluid, but also to study the long-term functional consequences in the context of pathology such as adult-onset obesity or inflammation.”

2) Evidence that Rac1/Rho signaling is involved in vivo: The authors observe a reduction of Claudin5 expression in LEC junctions after blocking EphrinB2 function, but Claudin5 knockout did not mimic the junctional defects seen in Efnb2 or Ephb4 mutants. Hence, additional mechanisms must be at play. The authors examine Rac1/Rho-mediated cytoskeletal organization. The cell culture work is nicely done and shows that Rock inhibition can rescue junctional and permeability defects induced by EphrinB2 function blocking antibodies. However, whether this accounts for the in vivo phenotype is not shown. Can pharmacological Rock inhibition rescue the mutant collecting junction phenotype? It should be possible to directly inject and determine if the leaky collecting vessels form improved zipper junctions. In the case that this further analysis cannot be accomplished, we request text revisions to acknowledge this caveat.

We agree and have attempted to rescue the collecting vessel phenotype in the *Efnb2* mutant mice by pharmacological ROCK inhibition. We first administered *Efnb2^flox/flox^;Prox1-CreER^T2^* litters with a single intraperitoneal injection of 4-OHT, followed by 4 consecutive intraperitoneal injections of Y27632 at P4, P6, P8 and P10 to cover the period of 7 days during which we found junctional defects to develop. We found that under these conditions, chylothorax formation and junctional defects were not prevented in Y27632-treated P11 *Efnb2^flox/flox^;Prox1-CreER^T2^* mice. It is possible that administration at 48 h intervals is not sufficient to provide a constant level of ROCK inhibition, considering that Y27632 has a half-life of 12-16 h (Mertsch et al., 2013). However, to achieve a complete long term ROCK inhibition would not be desirable during early postnatal development as it is likely to cause dysregulation of other pathways important for organ development.

To acknowledge the limitation of our data regarding the role of Rho signaling in the *Efnb2* mutant phenotype in vivo, we have revised the text in the Discussion as follows:

“In agreement with previous findings in HUVECs (Abengozar et al., 2012), we found that loss of EphrinB2/EphB4 signalling in primary LECs led to an increase in actin stress fibers that was attributed to increased Rho activity. Although in vivo validation remains to be done, our findings support a concept that continuous basal activation of EphrinB2/EphB4 signalling provides a homeostatic control of Rac1/Rho-mediated cytoskeletal contractility to regulate LEC junction integrity.”

We have also changed the wording throughout the text to ensure that findings related to the role of Rho signaling refer to in vitro experiments.

3) Antibody validation and comparison: have the authors validated the specificity and/or selectivity of the B11 function blocking antibody? For instance, through a comparison of junctional changes in siCTRL and siEFNB2 transfected cells, or phosphorylation of the EphB4 receptor, to make sure that the observed endothelial changes in response to B11 are indeed due to altered EphrinB2-EphB4 signaling. The authors state in the text (subsection “Selective reduction of junctional CLDN5 upon loss of basal EphrinB2/EphB4 signalling”, last paragraph) that the junctional defects with antibody incubation become more pronounced over time, but do not show the data. How do the LECs and the junctions look like after EphrinB2 and EphB4 siRNA knock down?

The B11 EphrinB2 blocking antibody has been extensively characterized by our collaborator Jorge L. Martínez-Torrecuadrada (who is also co-author of the current study), and validated by others (Krusche et al., 2016; Lennon et al., 2019). Dr. Martínez-Torrecuadrada’s group has previously demonstrated that EphrinB2-mediated phosphorylation of the EPHB4 receptor is prevented by pre-incubation with the B11 blocking antibody (Abengozar et al., 2012; Kwak et al., 2016). To provide further support for the role of EphrinB2 in regulating LEC junctions, and following the reviewers’ suggestion, we additionally show that siRNA-mediated *EFNB2* knock-down leads to similar disruption of VE-cadherin^+^ LEC junctions and downregulation of CLDN5 than observed after B11 treatment (new Figure 4—figure supplement 2A, B). We also provide the data showing more pronounced junctional disruption and reduction of CLDN5 upon extended (16 h) time of B11 incubation (Figure 4—figure supplement 1).